# Privacy Auditing with One (1) Training Run

**Thomas Steinke**[*]
Google DeepMind
steinke@google.com

**Milad Nasr**[*]
Google DeepMind
srxzr@google.com

**Matthew Jagielski**[*]
Google DeepMind
jagielski@google.com

## Abstract

We propose a scheme for auditing differentially private machine learning systems with a single training run. This exploits the parallelism of being able to add or remove multiple training examples independently. We analyze this using the connection between differential privacy and statistical generalization, which avoids the cost of group privacy. Our auditing scheme requires minimal assumptions about the algorithm and can be applied in the black-box or white-box setting. We demonstrate the effectiveness of our framework by applying it to DP-SGD, where we can achieve meaningful empirical privacy lower bounds by training only *one model*. In contrast, standard methods would require training hundreds of models.

## 1 Introduction

Differential privacy (DP) [DMNS06] provides a quantifiable privacy guarantee by ensuring that no person's data significantly affects the probability of any outcome. Formally, a randomized algorithm $M$ satisfies $(\varepsilon, \delta)$-DP if, for any pair of inputs $x, x'$ differing only by the addition or removal of one person's data and any measurable $S$, we have

$$\mathbb{P}\left[M(x) \in S\right] \le e^{\varepsilon} \cdot \mathbb{P}\left[M(x') \in S\right] + \delta. \tag{1}$$

A DP algorithm is accompanied by a mathematical proof giving an *upper bound* on the privacy parameters $\varepsilon$ and $\delta$. In contrast, a *privacy audit* provides an empirical *lower bound* on the privacy parameters. Privacy audits allow us to assess the tightness of the mathematical analysis [JUO20; NHSBTJCT23] or, if the lower and upper bounds are contradictory, to detect errors in the analysis or in the algorithm's implementation [DWWZK18; BGDCTV18; TTSSJC22].

Typically, privacy audits obtain a lower bound on the privacy parameters directly from the DP definition (1). That is, we construct a pair of inputs $x, x'$ and a set of outcomes $S$ and we estimate the probabilities $\mathbb{P}\left[M(x) \in S\right]$ and $\mathbb{P}\left[M(x') \in S\right]$. However, estimating these probabilities requires running $M$ hundreds of times. This approach to privacy auditing is computationally expensive, which raises the question *Can we perform privacy auditing using a single run of the algorithm M?*

### 1.1 Our Contributions

**Our approach:** The DP definition (1) considers adding or removing a single person's data to or from the dataset. We consider multiple people's data and the dataset independently includes or excludes each person's data point. Our analysis exploits the parallelism of multiple independent data points in a single run of the algorithm in lieu of multiple independent runs. This approach is commonly used as an *unproven* heuristic in prior work [MEMPST21; ZBWTSRPNK22].

Our auditing procedure operates as follows. We identify $m$ data points (i.e., training examples or "canaries") to either include or exclude and we flip $m$ independent unbiased coins to decide which of them to include or exclude. We then run the algorithm on the randomly selected dataset. Based

---

[*]Reverse alphabetical author order. Full version: https://arxiv.org/abs/2305.08846

37th Conference on Neural Information Processing Systems (NeurIPS 2023).

on the output of the algorithm, the auditor "guesses" whether or not each data point was included or excluded (or it can abstain from guessing for some data points). We obtain a lower bound on the privacy parameters from the fraction of guesses that were correct.

Intuitively, if the algorithm is $(\varepsilon, 0)$-DP, then the auditor can correctly guess each inclusion/exclusion coin flip with probability $\leq \frac{e^\varepsilon}{e^\varepsilon + 1}$. That is, the highest possible accuracy is attained by randomized response [War65]. Thus DP implies a high-probability upper bound on the fraction of correct guesses and, conversely, the fraction of correct guesses implies a high-probability lower bound on the privacy parameters.

**Our analysis:** Naïvely, analyzing the addition or removal of multiple data elements would rely on group privacy; but this does not exploit the fact that the data items were included or excluded independently. Instead, we leverage the connection between DP and generalization [DFHPRR15b; DFHPRR15a; BNSSSU16; RRST16; JLNRSMS19; SZ20]. Our main theoretical contribution is an improved analysis of this connection that is tailored to yield nearly tight bounds in our setting.

Informally, if we run a DP algorithm on i.i.d. samples from some distribution, then, conditioned on the output of the algorithm, the samples are still "close" to being i.i.d. samples from that distribution. There is some technicality in making this precise, but, roughly speaking, we show that including or excluding $m$ data points independently for one run is essentially as good as having $m$ independent runs (as long as $\delta$ is small).

**Our results:** As an application of our new auditing framework, we audit DP-SGD training on a WideResNet model, trained on the CIFAR10 dataset across multiple configurations. Our approach successfully achieves an empirical lower bound of $\varepsilon \geq 1.8$, compared to a theoretical upper bound of $\varepsilon \leq 4$ in the white-box setting. The $m$ examples we insert for auditing (known in the literature as "canaries") do not significantly impact the accuracy of the final model (less than a $5\%$ decrease in accuracy) and our procedure only requires a single end-to-end training run. Such results were previously unattainable in the setting where only one model could be trained.

## 2 Our Auditing Procedure

---
**Algorithm 1** Auditor with One Training Run

---
1: **Data:** $x \in \mathcal{X}^n$ consisting of $m$ auditing examples (a.k.a. canaries) $x_1, \cdots, x_m$ and $n - m$ non-auditing examples $x_{m+1}, \cdots, x_n$.
2: **Parameters:** Algorithm to audit $\mathcal{A}$, number of examples to randomize $m$, number of positive $k_+$ and negative $k_-$ guesses.
3: For $i \in [m]$, sample $S_i \in \{-1, +1\}$ uniformly and independently. Set $S_i = 1$ for all $i \in [n] \setminus [m]$.
4: Partition $x$ into $x_{\text{IN}} \in \mathcal{X}^{n_{\text{IN}}}$ and $x_{\text{OUT}} \in \mathcal{X}^{n_{\text{OUT}}}$ according to $S$, where $n_{\text{IN}} + n_{\text{OUT}} = n$. Namely, if $S_i = 1$, then $x_i$ is in $x_{\text{IN}}$; and, if $S_i = -1$, then $x_i$ is in $x_{\text{OUT}}$.
5: Run $\mathcal{A}$ on input $x_{\text{IN}}$ with appropriate parameters, outputting $w$.
6: Compute the vector of scores $Y = (\text{SCORE}(x_i, w) : i \in [m]) \in \mathbb{R}^m$.
7: Sort the scores $Y$. Let $T \in \{-1, 0, +1\}^m$ be $+1$ for the largest $k_+$ scores and $-1$ for the smallest $k_-$ scores. (I.e., $T \in \{-1, 0, +1\}^m$ maximizes $\sum_i^m T_i \cdot Y_i$ subject to $\sum_i^m |T_i| = k_+ + k_-$ and $\sum_i^m T_i = k_+ - k_-$.)
8: **Return:** $S \in \{-1, +1\}^m$ indicating the true selection and the guesses $T \in \{-1, 0, +1\}^m$.

---

We present our auditing procedure in Algorithm 1. We independently include each of the first $m$ examples with 50% probability and exclude it otherwise.[2] When applied to DP-SGD, our approach is applicable to both "white-box" auditing, where the adversary can access to all intermediate values of the model weights (as in the federated learning setting), and "black-box" auditing, where the adversary only sees the final model weights (or can only query the final model, as in the centralized setting). In both cases we simply compute a "score" for each example and "guess" whether the example is included or excluded based on these scores. Specifically, we guess that the examples with the $k_+$ highest scores are included and the examples with the $k_-$ lowest scores are excluded,

---
[2]We consider the add/remove notion of DP, so changing one $S_i$ results in neighbouring inputs $x_{\text{IN}}$. Our methods readily extend to the replacement notion of DP; see the full version.

and we abstain from guessing for the remaining $m - k_+ - k_-$ auditing examples; the setting of these parameters will depend on the application.

Note that we only randomize the first $m$ examples $x_1, \cdots, x_m$ (which we refer to as "auditing examples" or "canaries"); the last $n - m$ examples $x_{m+1}, \cdots, x_n$ are always included and, thus, we do not make any guesses about them. To get the strongest auditing results we would set $m = n$, but we usually want to set $m < n$. For example, computing the score of all $n$ examples may be computationally prohibitive, so we only compute the scores of $m$ examples. We may wish to artificially construct $m$ examples to be easy to identify (i.e., canaries), but also include $n - m$ "real" examples to ensure that $\mathcal{A}$ still produces a useful model. (I.e., having more training examples improves the performance of the model.)

The score function is arbitrary and will depend on the application. For black-box auditing, we use the loss of the final model $w$ on the example $x_i$ – i.e., $\text{SCORE}(x_i, w) = -\text{loss}(w, x_i)$. For white-box auditing, $\text{SCORE}(x_i, w) = \sum_t^\ell \left\langle w^{t-1} - w^t, \nabla_{w^{t-1}} \text{loss}(w^{t-1}, x_i) \right\rangle$ is the sum of the inner products of updates with the (clipped) gradients of the loss on the example.

Intuitively, the vector of scores $Y$ should be correlated with the true selection $S$, but too strong a correlation would violate DP. This is the basis of our audit. Specifically, the auditor computes $T$ from $Y$ which is a "guess" at $S$. By the postprocessing property of DP, the guesses $T$ are a differentially private function of the true $S$, which means that they cannot be too accurate.

**Why abstain from guessing?** The guesses are binary ($T_i = 1$ for IN, $T_i = -1$ for OUT). Abstaining ($T_i = 0$) allows us to capture uncertainty. That is, on some examples we will not be sure whether they are IN or OUT. By abstaining on these uncertain examples, we can ensure higher accuracy for the remaining guesses. This yields better auditing results in practice.

## 3 Theoretical Analysis

To obtain a lower bound on the DP parameters we show that DP implies a high-probability upper bound on the number of correct guesses $W := \sum_i^m \max\{0, T_i \cdot S_i\}$ of our auditing procedure (Algorithm 1). Note that $\max\{0, T_i \cdot S_i\}$ is simply 1 if the guess was correct – i.e., $T_i = S_i$ – and 0 otherwise. The observed value of $W$ then yields a high-probability lower bound on the DP parameters. To be more precise, we have the following guarantee.

**Theorem 3.1** (Main Result). *Let $(S, T) \in \{-1, +1\}^m \times \{-1, 0, +1\}^m$ be the output of Algorithm 1. Assume the algorithm to audit $\mathcal{A}$ satisfies $(\varepsilon, \delta)$-DP. Let $r := k_+ + k_- = \|T\|_1$ be the number of guesses. Then, for all $v \in \mathbb{R}$,*

$$\mathbb{P}\left[\sum_i^m \max\{0, T_i \cdot S_i\} \geq v\right] \leq \mathop{\mathbb{P}}_{\check{W} \leftarrow \text{Binomial}\left(r, \frac{e^\varepsilon}{e^\varepsilon + 1}\right)}\left[\check{W} \geq v\right] + \delta \cdot m \cdot \alpha, \tag{2}$$

*where*

$$\alpha = \max_{i \in [m]} \frac{2}{i} \cdot \mathop{\mathbb{P}}_{\check{W} \leftarrow \text{Binomial}\left(r, \frac{e^\varepsilon}{e^\varepsilon + 1}\right)}\left[v > \check{W} \geq v - i\right]. \tag{3}$$

If we ignore $\delta$ for the moment, Theorem 3.1 says that the number of correct guesses is stochastically dominated by $\text{Binomial}\left(r, \frac{e^\varepsilon}{e^\varepsilon + 1}\right)$, where $r = k_+ + k_-$ is the total number of guesses. This binomial distribution is precisely the distribution of correct guesses we would get if $T$ was obtained by independently performing $(\varepsilon, 0)$-DP randomized response on $r$ bits of $S$. In other words, the theorem says that $(\varepsilon, 0)$-DP randomized response is exactly the worst-case algorithm in terms of the number of correct guesses. In particular, this means the theorem is tight (when $\delta = 0$).

The binomial distribution is well-concentrated. In particular, for all $\beta \in (0, 1)$, we have

$$\mathop{\mathbb{P}}_{\check{W} \leftarrow \text{Binomial}\left(r, \frac{e^\varepsilon}{e^\varepsilon + 1}\right)}\left[\check{W} \geq \underbrace{\frac{r \cdot e^\varepsilon}{e^\varepsilon + 1} + \sqrt{\frac{1}{2} \cdot r \cdot \log(1/\beta)}}_{=v}\right] \leq \beta. \tag{4}$$

There is an additional $O(\delta)$ term in the guarantee (2). The exact expression (3) is somewhat complex. It is always $\leq 2m\delta$, since $\alpha \leq 2$, but it is much smaller than this for reasonable parameter values. In particular, for $v$ as in Equation 4 with $\beta \leq 1/r^4$, we have $\alpha \leq O(1/r)$.

Theorem 3.1 gives us a hypothesis test: If $\mathcal{A}$ is $(\varepsilon, \delta)$-DP, then the number of correct guesses $W$ is $\leq \frac{r \cdot e^\varepsilon}{e^\varepsilon + 1} + O(\sqrt{r})$ with high probability. Thus, if the observed number of correct guesses $v$ is larger than this, we can reject the hypothesis that $\mathcal{A}$ satisfies $(\varepsilon, \delta)$-DP. We can convert this hypothesis test into a confidence interval (i.e., a lower bound on $\varepsilon$) by finding the largest $\varepsilon$ that we can reject at a desired level of confidence.

**Proof of Theorem 3.1:** Due to space limitations, the proof is deferred to the full version. But we outline the main ideas: First note that $S \in \{-1, +1\}^m$ is uniform and $T$ is a DP function of $S$. Consider the distribution of $S$ conditioned on $T = t$. If $\mathcal{A}$ is pure $(\varepsilon, 0)$-DP, then we can easily analyze the conditional distribution using Bayes' law [DFHPRR15a; RRST16; JLNRSMS19] to conclude that each guess has probability $\leq \frac{e^\varepsilon}{e^\varepsilon + 1}$ of being correct. Furthermore, this holds even if we condition on the other guesses, which allows us to inductively prove that the number of correct guesses is stochastically dominated by $\text{Binomial}\left(r, \frac{e^\varepsilon}{e^\varepsilon + 1}\right)$. Handling approximate DP ($\delta > 0$) introduces additional complexity – some outputs $T$ are "bad" in the sense that the conditional distribution of $S_i$ could be arbitrary. Fortunately, such bad outputs are rare [KS14]. What we can show is that the number of correct guesses is stochastically dominated by $\check{W} + F(T)$, where $\check{W} \leftarrow \text{Binomial}\left(r, \frac{e^\varepsilon}{e^\varepsilon + 1}\right)$ is as before and $F(T) \in \{0, 1, \cdots, m\}$ indicates how many of these bad events happened. We do not know the exact distribution of $F(T)$, but we do know $\mathbb{E}[F(T)] \leq 2m\delta$, which suffices to prove our result. Equation 3 comes from looking for the worst-case $F(T)$; essentially the worst case is $\mathbb{P}[F(T) = i] = 2m\delta/i$ and $\mathbb{P}[F(T) = 0] = 1 - 2m\delta/i$ for some $i \in [m]$.

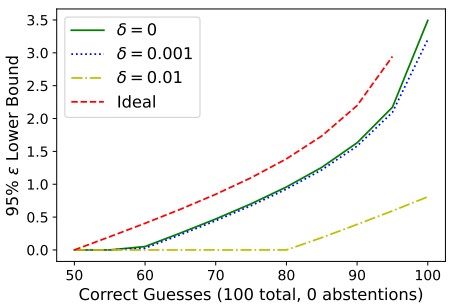

Figure 1: Lower bound on the privacy parameter $\varepsilon$ given by Theorem 3.1 with 95% confidence as the number of correct guesses changes. The total number of examples and guesses is 100. For comparison, we plot the `ideal` $\varepsilon$ that gives $100 \cdot \frac{e^\varepsilon}{e^\varepsilon + 1}$ correct guesses.

Figure 2: Lower bound on the privacy parameter $\varepsilon$ given by Theorem 3.1 with 95% confidence as the number of correct guesses changes. The total number of examples and guesses is 1000 (with no abstentions). Here we plot the ideal $\varepsilon$ on the horizontal axis, so that the number of correct guesses is $1000 \cdot \frac{e^\varepsilon}{e^\varepsilon + 1}$.

Our theoretical analysis also gives novel implications for generalization from DP, improving on the results of Jung, Ligett, Neel, Roth, Sharifi-Malvajerdi, and Shenfeld [JLNRSMS19]. In addition, we obtain bounds on the mutual information $I(S; M(S)) \leq \frac{1}{8}\varepsilon^2 m \log e + \delta m \log 2$, where $S \in \{0, 1\}^m$ is uniformly random and $M : \{0, 1\}^m \to \mathcal{Y}$ satisfies $(\varepsilon, \delta)$-DP, which improves on the results of Steinke and Zakynthinou [SZ20]. See the full version for more details.

## 4 Experiments

**Experiment Setup** Our contributions are focused on improved analysis of an existing privacy attack, and are therefore orthogonal to the design of an attack. As a result, we rely on the experimental setup of the recent auditing procedure of Nasr, Hayes, Steinke, Balle, Tramèr, Jagielski, Carlini, and Terzis [NHSBTJCT23].

We run DP-SGD on the CIFAR-10 dataset with Wide ResNet (WRN-16) [ZK16], following the experimental setup of Nasr et al. [NHSBTJCT23]. Our experiments reach 76% test accuracy at $(\varepsilon = 8, \delta = 10^{-5})$-DP, which is comparable with the state-of-the-art [DBHSB22]. Unless specified

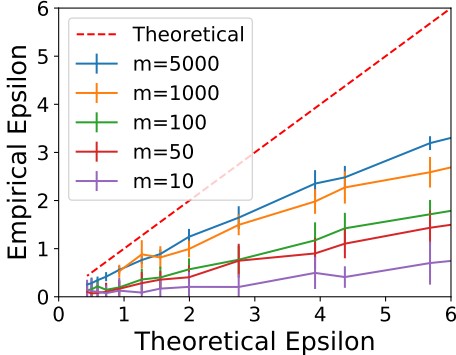 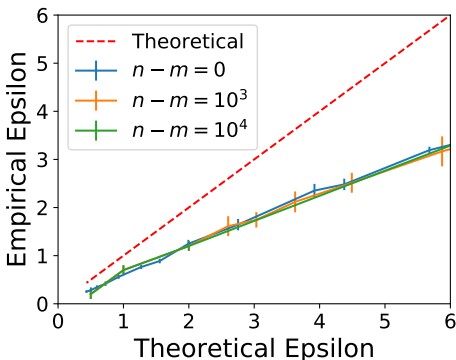

Figure 3: Effect of the number of auditing examples $(m)$ in the white-box setting. By increasing the number of the auditing examples we are able to achieve tighter empirical lower bounds.

Figure 4: Effect of the number of additional examples $(n - m)$ in the white-box setting. Importantly, adding additional examples does not impact the auditing results in the white-box setting.

otherwise, all lower bounds are presented with $95\%$ confidence. Following prior work, we refer to the setting where the adversary has access to all intermediate steps as "white-box" and when the adversary can only see the last iteration as "black-box." We experiment with both settings.

Algorithm 1 summarizes our approach for auditing DP-SGD. The results are converted into lower bounds on the privacy parameters using Theorem 3.1.

We also experiment with both the gradient and input attacks proposed by Nasr et al. [NHSBTJCT23]. In particular, for the gradient attack we use the strongest attack they proposed – the "Dirac canary" approach – which sets all gradients to zero except at a single random index. In our setting where we need to create multiple auditing examples (canaries) we make sure the indices selected in our experiments do not have any repetitions. To compute the score for gradient space attacks, we use the dot product between the gradient update and auditing gradient. When auditing in input space, we leverage two different types of injected examples as:

1. **Mislabeled example**: We select a random subset of the test set and randomly relabel them (ensuring the new label is not the same as the original label).

2. **In-distribution example**: We select a random subset of the test set.

For input space audits, we use the loss of the input example as the score. In our experiments we report the attack with the highest lower bound.

In our experiments, we evaluate different values of $k_+$ and $k_-$ and report the best auditing results. Since changing the number of guesses $r = k_+ + k_-$ versus number of abstentions $m - r$ is potentially testing multiple hypothesis on the same data, we should reduce the reported confidence value of our results. However, common practice is to simply ignore this minor issue [ZBWTSRPNK22; MSS22]. Fortunately, our main theorem can be extended to account for the dynamic choice of the number of guesses (see Corollary 5.8 in the full version).

### 4.1 Gradient Space attacks

We start with the strongest attack: We assume white-box access – i.e., the auditor sees all intermediate iterates of DP-SGD (which is a reasonable assumption in the federated learning setting) – and that the auditor can insert examples with arbitrary gradients into the training procedure. First, we evaluate the effect of the number of the auditing example on the tightness. Figure 3 demonstrates that as the number of examples increases, the auditing becomes tighter. However, the impact of the additional examples eventually diminishes. Intriguingly, adding more non-auditing training examples (resulting in a larger $n$ compared to $m$) does not seem to influence the tightness of the auditing, as depicted in Figure 4. This can be primarily due to the fact that gradient attacks proposed in prior studies can generate near-worst-case datasets, irrespective of the presence of other data points.

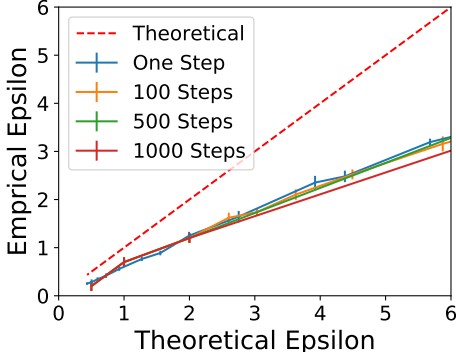 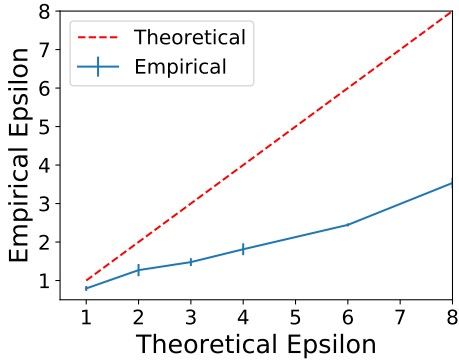

Figure 5: Effect of number of iterations in the white-box setting. Increasing the number of the steps (while keeping the same overall privacy by increasing the added noise) will not effect the auditing results.

Figure 6: Auditing CIFAR10 SoTA in white-box setting using gradient attacks. Our auditing framework can achieve meaningful empirical privacy lower bounds for SoTA models.

Another parameter that might affect the auditing results is the number of iterations $\ell$ in the DP-SGD algorithm. As shown in Figure 5 we compare the extreme setting of having one iteration to multiple iterations and we do not observe any significant difference in the auditing when auditing for the equivalent privacy guarantees (by increasing the noise). The results confirm the tightness of composition and that the number of iterations does not have significant effect on auditing in white-box setting.

Now we directly use the parameters used in the training CIFAR10 models. Figure 6 summarizes results for the CIFAR10 models. We used $m = 5000$ and all of the training dataset from CIFAR10 ($n = 50,000$) for the attack. We were able to achieve $76\%$ accuracy for $\varepsilon = 8$ ($\delta = 10^{-5}$, compared to $78\%$ when not auditing). We are able to achieve an empirical lower bound of $0.7, 1.2, 1.8, 3.5$ for theoretical epsilon of $1, 2, 4, 8$ respectively. While our results are not as tight as the prior works, we only require a single run of training which is not possible using the existing techniques. In the era of exponentially expanding machine learning models, the computational and financial costs of training these colossal architectures even once are significant. Expecting any individual or entity to shoulder the burden of training such models thousands of times for the sake of auditing or experimental purposes is both unrealistic and economically infeasible. Our method offers a unique advantage by facilitating the auditing of these models, allowing for an estimation of privacy leakage in a white-box setting without significantly affecting performance.

## 4.2 Input Space Attacks

Now we evaluate the effect of input space attacks in the black-box setting. In this attack, the auditor can only insert actual images into the training procedure and cannot control any of the aspects of the training. Then, the adversary can observe the final model as mentioned in Algorithm 1. This is the weakest attack setting.

For simplicity we start with the setting where $m = n$; in other words, all of the examples used to train the model are randomly included or excluded and can be used for auditing. Figure 7 illustrates the result of this setting. As we see from the figure, unlike the white-box attack we do not observe a monotonic relationship between the number of auditing examples and the tightness of the auditing. Intuitively, when the number of auditing examples are low then we do not have enough observations to have high confidence lower bounds for epsilon. On the other hand, when the number of auditing examples are high, the model does not have enough capacity to "memorize" all of the auditing examples which reduces the tightness of the auditing. However, this can be improved by designing better black-box attacks which we reiterate in the next section.

We also evaluate the effect of adding additional training data to the auditing in Figure 8. We see that adding superfluous training data significantly reduces the effectiveness of auditing. The observed reduction in auditing effectiveness with the addition of more training data could be attributed to

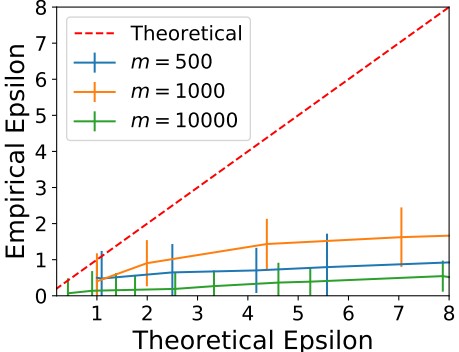 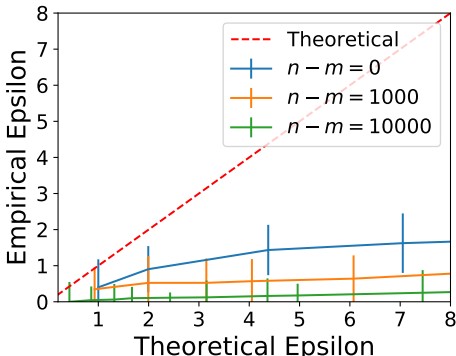

Figure 7: Effect of the number of auditing examples ($m$) in the black-box setting. Black-box auditing is very sensitive to the number of auditing examples.

Figure 8: Effect of the number of additional example on auditing ($n-m$) in the black-box setting. By increasing the number of additional examples, the auditing results get significantly looser.

several factors. One interpretation could be that the theoretical privacy analysis in a black-box setting tends to be considerably more loose when the adversary is constrained to this setting. This could potentially result in an overestimation of the privacy bounds. Conversely, it is also plausible that the results are due to the weak black-box attacks and can be improved in the future.

## 5 Related Work

The goal of privacy auditing is to empirically estimate the privacy provided by an algorithm, typically to accompany a formal privacy guarantee. Early work on auditing has often been motivated by trying to identify bugs in the implementations of differentially private data analysis algorithms [DWWZK18; BGDCTV18].

Techniques for auditing differentially private machine learning typically rely on conducting some form of membership inference attack [SSSS17];[3] these attacks are designed to detect the presence or absence of an individual example in the training set. Essentially, a membership inference attack which achieves some true positive rate (TPR) and false positive rate (FPR) gives a lower bound on the privacy parameter $\varepsilon \geq \log_e(\text{TPR}/\text{FPR})$ (after ensuring statistical validity of the TPR and FPR estimates).

Jayaraman and Evans [JE19] use standard membership inference attacks to evaluate different privacy analysis algorithms. Jagielski, Ullman, and Oprea [JUO20] consider inferring membership of worst-case "poisoning" examples to conduct stronger membership inference attacks and understand the tightness of privacy analysis. Nasr, Song, Thakurta, Papernot, and Carlini [NSTPC21] measure the tightness of privacy analysis under a variety of threat models, including showing that the DP-SGD analysis is tight in the threat model assumed by the standard DP-SGD analysis.

Improvements to auditing have been made in a variety of directions. For example, Nasr, Hayes, Steinke, Balle, Tramèr, Jagielski, Carlini, and Terzis [NHSBTJCT23] and Maddock, Sablayrolles, and Stock [MSS22] take advantage of the iterative nature of DP-SGD, auditing individual steps to understand privacy of the end-to-end algorithm. Improvements have also been made to the basic statistical techniques for estimating the $\varepsilon$ parameter, for example by using Log-Katz confidence intervals [LMFLZWRFT22], Bayesian techniques [ZBWTSRPNK22], or auditing algorithms in different privacy definitions [NHSBTJCT23].

Andrew, Kairouz, Oh, Oprea, McMahan, and Suriyakumar [AKOOMS23] build on the observation that, when performing membership inference, analyzing the case where the data is not included does

---

[3]Shokri, Stronati, Song, and Shmatikov [SSSS17] coined the term "membership inference attack" and were the first to apply such attacks to machine learning systems. However, similar attacks were developed for applications to genetic data [HSRDTMPSNC08; SOJH09; DSSUV15] and in cryptography [BS98; Tar08].

not require re-running the algorithm; instead we can re-sample the excluded data point; if the data points are i.i.d. from a nice distribution, this permits closed-form analysis of the excluded case. This gives a method for estimating the privacy parameters of an algorithm in a single run, but it does not guarantee that the estimate is either a lower or upper bound. In a similar vein, independent and concurrent work by Pillutla, Andrew, Kairouz, McMahan, Oprea, and Oh [PAKMOO23] provides a rigorous privacy auditing scheme that re-uses training runs to improve efficiency. Rather than consider a single pair of inputs differing by a single example, they consider multiple pairs of neighbouring inputs, but these pairs are overlapping in the sense that a single input dataset may appear in multiple pairs. Thus each training run can be re-used to estimate the privacy parameters of multiple pairs simultaneously or, rather, a single lower bound on the privacy parameters is computed over a (random) combination of the pairs. This is significantly more efficient than the naïve approach, but still requires more than one run.

A recent heuristic proposed to improve the efficiency of auditing is performing membership inference on multiple examples simultaneously. This heuristic was proposed by Malek Esmaeili, Mironov, Prasad, Shilov, and Tramer [MEMPST21], and evaluated more rigorously by Zanella-Béguelin, Wutschitz, Tople, Salem, Rühle, Paverd, Naseri, and Köpf [ZBWTSRPNK22]. However, this heuristic is not theoretically justified, as the TPR and FPR estimates are not based on independent samples. In our work, we provide a proof of the validity of this heuristic. In fact, with this proof, we show for the first time that standard membership inference attacks, which attack multiple examples per training run, can be used for auditing analysis; prior work using these attacks must make an independence assumption. As a result, auditing can take advantage of progress in the membership inference field [CCNSTT22; WBKBGGG22].

Our theoretical analysis builds on the connection between DP and generalization [DFHPRR15b; DFHPRR15a; BNSSSU16; RRST16; JLNRSMS19; SZ20]. Our approach can be viewed as a contrapositive to "privacy implies generalization." That is, "failure to generalize implies non-privacy." Our auditing scheme can also be viewed as a "reconstruction attack" [DN03; DSSU17]. That is, the auditor's guesses $T$ can be viewed as a reconstruction of the private coins $S$. And it is known that DP prevents reconstruction [De12]

# 6 Discussion

Our main contribution is showing that we can audit the differential privacy guarantees of an algorithm with a single run. In contrast, prior methods require hundreds – if not thousands – of runs, which is computationally prohibitive for all but the simplest algorithms and machine learning models. Our experimental results demonstrate that in practical settings our methods are able to give meaningful lower bounds on the privacy parameter $\varepsilon$.

However, while we win on computational efficiency, we lose on tightness of our lower bounds. We now illustrate the limitations of our approach and discuss the extent to which this is inherent, and what lessons we can learn.

But, first, we illustrate that our method can give tight lower bounds. In Figure 9, we consider an idealized setting where the number of guesses changes and the fraction that are correct is fixed at $\frac{e^\varepsilon}{e^\varepsilon+1}$ for $\varepsilon = 4$ – i.e., 98.2% of guesses are correct.[4] This is the maximum expected fraction of correct guesses compatible with $(4, 0)$-DP. In this setting the lower bound on $\varepsilon$ does indeed come close to 4. With 10,000 guesses we get $\varepsilon \geq 3.87$ with 95% confidence.

Note that the lower bound in Figure 9 improves as we increase the number of guesses. This is simply accounting for sampling error – to get a lower bound with 95% confidence, we must underestimate to account for the fact that the number of correct guesses may have been inflated by chance. As we get more guesses, the relative size of chance deviations reduces.

**Limitations:** Next we consider a different idealized setting – one that is arguably more realistic – where our method does *not* give tight lower bounds. Suppose $S_i \in \{-1, +1\}$ indicates whether example $i \in [n]$ is included or excluded. In Figure 10, we consider Gaussian noise addition. That is, we release a sample from $\mathcal{N}(S_i, 4)$. (In contrast, Figure 9 considers randomized response on $S_i$.) A tight analysis of the Gaussian mechanism [BW18, Theorem 8] gives an upper bound of

---

[4]The number of correct guesses is rounded down to an integer (which results in the lines being jagged). There are no abstentions.

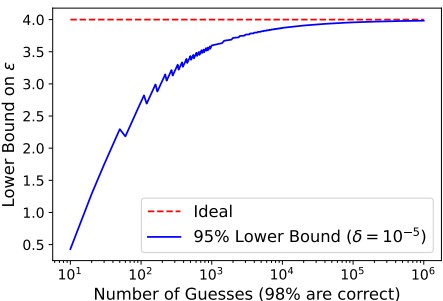

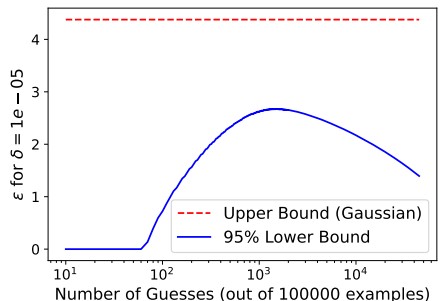

Figure 9: Comparison of upper and lower bounds for idealized setting with varying number of guesses. The fraction of correct guesses is always $\frac{e^\varepsilon}{e^\varepsilon+1} \approx 0.982$ for $\varepsilon = 4$.

Figure 10: Comparison of upper and lower bounds for idealized setting with varying number of guesses. For each example $i \in [m]$, we release $S_i + \xi_i$, where $\xi_i \leftarrow \mathcal{N}(0,4)$ and $S_i \in \{-1,+1\}$ is independently uniformly random and indicates whether the sample is included/excluded. For the upper bound, we compute the exact $(4.38, 10^{-5})$-DP guarantee for the Gaussian mechanism. For the lower bound, we plot the bound of Theorem 3.1 with 95% confidence for varying numbers of guesses $r$. Total of $m = 100,000$ randomized examples; we guess $T_i = +1$ for the largest $r/2$ scores and $T_i = -1$ for the smallest $r/2$ scores; we guess $T_i = 0$ for the remaining $m - r$ examples. The number of correct guesses is set to $\lceil r \cdot \mathbb{P}[S_i = +1|S_i + \xi_i > c] \rceil$, where $c$ is a threshold such that $\mathbb{P}[S_i + \xi_i > c] = \frac{r}{2m}$.

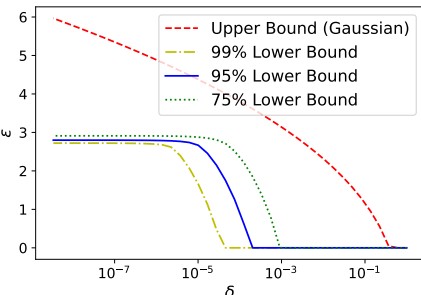

Figure 11: Same setting as Figure 10 with varying $\delta$ and confidence, but number of guesses fixed to $r =$ 1,500 of which 1,429 are correct out of $m =$ 100,000 auditing examples.

$(4.38, 10^{-5})$-DP. Unlike for randomized response, abstentions matter here. We consider 100,000 examples, each of which has a score sampled from $\mathcal{N}(S_i, 4)$, where $S_i \in \{-1,+1\}$ is uniformly random. We pick the largest $r/2$ scores and guess $S_i = +1$. Similarly we guess $S_i = -1$ for the smallest $r/2$ scores. We abstain for the remaining $100,000 - r$ examples. If we make more guesses (i.e., increase $r$), then we start making guesses on examples which are less "in the tails" of each distribution, which provide less signal for the attack, making the attack accuracy go down along with our lower bound. We must trade off between more guesses being less accurate on average and more guesses having smaller relative sampling error.

In Figure 10, the highest value of the lower bound is $\varepsilon \geq 2.675$ for $\delta = 10^{-5}$, which is attained by 1439 correct guesses out of 1510. In contrast, the upper bound is $\varepsilon = 4.38$ for $\delta = 10^{-5}$. To get a matching upper bound of $\varepsilon \leq 2.675$ we would need to set $\delta = 0.0039334$. In other words, the gap between the upper and lower bounds is a factor of $393\times$ in $\delta$.

Figure 11 considers the same idealized setting as Figure 10, but we fix the number of guesses to 1,500 out of 100,000 (of which 1,429 are correct); instead we vary $\delta$ and the confidence level.

**Are these limitations inherent?** Figures 10 & 11 illustrate the limitations of our approach. They also hint at the cause: The number of guesses versus abstentions, the $\delta$ parameter, and the confidence all have a large effect on the tightness of our lower bound.

Our theoretical analysis is fairly tight; there is little room to improve Theorem 3.1. We argue that the inherent problem is a mismatch between "realistic" DP algorithms and the "pathological" DP algorithms for which our analysis is nearly tight. This mismatch makes our lower bound much more sensitive to $\delta$ than it "should" be.

To be concrete about what we consider pathological, consider $M : \{-1,+1\}^m \rightarrow \{-1,0,+1\}^m$ defined by Algorithm 2. This algorithm satisfies $(\varepsilon,\delta)$-DP and makes $r$ guesses with $m-r$ abstentions. In the $X=1$ case, the expected fraction of correct guesses is $\frac{m\delta}{r\beta} + \left(1 - \frac{m\delta}{r\beta}\right) \cdot \frac{e^\varepsilon}{e^\varepsilon+1}$. This is higher than the average fraction of correct guesses, but if we want confidence $1-\beta$ in our lower bound, we must consider this case, as $X=1$ happens with probability $\beta$.

Intuitively, the contribution from $\delta$ to the fraction of correct guesses should be negligible. However, we see that $\delta$ is multiplied by $m/r\beta$. That is to say, in the settings we consider, $\delta$ is multiplied by a factor on the order of $100\times$ or $1000\times$, which means $\delta = 10^{-5}$ makes a non-negligible contribution to the fraction of correct guesses.

---

**Algorithm 2** Pathological Algorithm

1: **Input:** $s \in \{-1,+1\}^m$
2: **Parameters:** $r \in [m]$, $\varepsilon, \delta \geq 0$, $\beta \in [0,1]$. Assume $0 < m\delta \leq r\beta$.
3: Select $U \subset [m]$ with $|U| = r$ uniformly random.
4: Set $T_i = 0$ for all $i \notin U$.
5: Sample $X \leftarrow \text{Bernoulli}(\beta)$.
6: **if** $X = 1$ **then**
7:     **for** $i \in U$ **do**
8:         Independently sample $T_i \in \{-1,+1\}$ with
$$\mathbb{P}\left[T_i = s_i\right] = \frac{m\delta}{r\beta} + \left(1 - \frac{m\delta}{r\beta}\right) \cdot \frac{e^\varepsilon}{e^\varepsilon+1}.$$
9:     **end for**
10: **else if** $X = 0$ **then**
11:     **for** $i \in U$ **do**
12:         Independently sample $T_i \in \{-1,+1\}$ with
$$\mathbb{P}\left[T_i = s_i\right] = \frac{e^\varepsilon}{e^\varepsilon+1}.$$
13:     **end for**
14: **end if**
15: **Output:** $T \in \{-1,0,+1\}^m$.

---

It is tempting to try to circumvent this problem by simply setting $\delta$ to be very small. However, as shown in Figure 11, the corresponding upper bound on $\varepsilon$ also increases as $\delta \rightarrow 0$.

Unfortunately, there is no obvious general way to rule out algorithms that behave like Algorithm 2. The fundamental issue is that the privacy losses of the $m$ examples are not independent; nor should we expect them to be, but they shouldn't be pathologically dependent either.

**Directions for further work:** Our work highlights several questions for further exploration:

- **Improved attacks:** Our experimental evaluation uses existing attack methods. Any improvements to membership inference attacks could be combined with our results to yield improved privacy auditing.

  One limitation of our attacks is that some examples may be "harder" than others and the scores we compute do not account for this. When we have many runs, we can account for the hardness of individual examples [CCNSTT22], but in our setting it is not obvious how to do this.

- **Algorithm-specific analyses:** Our methods are generic – they can be applied to essentially any DP algorithm. This is a strength, but there is also the possibility that we could obtain stronger results by exploiting the structure of specific algorithms. A natural example of such structure is the iterative nature of DP-SGD. That is, we can view one run of DP-SGD as the composition of multiple independent DP algorithms which are run sequentially.

- **Multiple runs & multiple examples:** Our method performs auditing by including or excluding multiple examples in a single training run, while most prior work performs multiple training runs with a single example example included or excluded. Can we get the best of both worlds? If we use multiple examples and multiple runs, we should be able to get tighter results with fewer runs.

- **Other measures of privacy:** Our theoretical analysis is tailored to the standard definition of differential privacy. But there are other definitions of differential privacy such as Rényi DP. And, in particular, many of the upper bounds are stated in this language. Hence it would make sense for the lower bounds also to be stated in this language.

- **Beyond lower bounds:** Privacy auditing produces empirical lower bounds on the privacy parameters. In contrast, mathematical analysis produces upper bounds. Both are necessarily conservative, which leaves a large gap between the upper and lower bounds. A natural question is to find some middle ground – an estimate which is neither a lower nor upper bound, but provides some meaningful estimate of the "true" privacy loss. However, it is unclear what kind of guarantee such an estimate should satisfy, or what interpretation the estimate should permit.

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
