# Privacy Auditing with One (1) Training Run

Thomas Steinke*        Milad Nasr*        Matthew Jagielski*

**Abstract**

We propose a scheme for auditing differentially private machine learning systems with a single training run. This exploits the parallelism of being able to add or remove multiple training examples independently. We analyze this using the connection between differential privacy and statistical generalization, which avoids the cost of group privacy. Our auditing scheme requires minimal assumptions about the algorithm and can be applied in the black-box or white-box setting.

## 1  Introduction

Differential privacy (DP) [DMNS06] provides a quantifiable privacy guarantee by ensuring that no person's data significantly affects the probability of any outcome. Formally, a randomized algorithm $M$ satisfies $(\varepsilon, \delta)$-DP if, for any pair of inputs $x, x'$ differing only by the addition or removal of one person's data and any measurable $S$, we have

$$\mathbb{P}\left[M(x) \in S\right] \le e^{\varepsilon} \cdot \mathbb{P}\left[M(x') \in S\right] + \delta. \tag{1}$$

A DP algorithm is accompanied by a mathematical proof giving an *upper bound* on the privacy parameters $\varepsilon$ and $\delta$. In contrast, a *privacy audit* provides an empirical *lower bound* on the privacy parameters. Privacy audits allow us to assess the tightness of the mathematical analysis [JUO20; NHSBTJCT23] or, if the lower and upper bounds are contradictory, to detect errors in the analysis or in the algorithm's implementation [TTSSJC22].

Typically, privacy audits obtain a lower bound on the privacy parameters directly from the DP definition (1). That is, we construct a pair of inputs $x, x'$ and a set of outcomes $S$ and we estimate the probabilities $\mathbb{P}\left[M(x) \in S\right]$ and $\mathbb{P}\left[M(x') \in S\right]$. However, estimating these probabilities requires running the algorithm $M$ hundreds of times. This approach to privacy auditing is computationally expensive, which raises the question

*Can we perform privacy auditing using a single run of the algorithm $M$?*

This is the question we address in our work.

---

*Google. {steinke,srxzr,jagielski}@google.com. Reverse alphabetical author order.

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

# 4 Background

We briefly review some standard background material. Readers may wish to skip to the next section and revisit this only if necessary.

## 4.1 Differential Privacy

They We recite the definitions of differential privacy and some relevant relaxations. For detailed background, see the tutorial by Vadhan [Vad17] or the textbook by Dwork and Roth [DR14].

**Definition 4.1** (Differential Privacy [DMNS06; DKMMN06]). *Let $M : \mathcal{X}^* \to \mathcal{Y}$ be a randomized algorithm, where $\mathcal{X}^* = \bigcup_{n \geq 0} \mathcal{X}^n$. We say $M$ is $(\varepsilon, \delta)$-differentially private $((\varepsilon, \delta)$-DP) if, for all $x, x' \in \mathcal{X}^*$ differing only by the addition or removal of one element, we have*

$$\forall S \subset \mathcal{Y} \qquad \mathbb{P}\left[M(x) \in S\right] \leq e^\varepsilon \cdot \mathbb{P}\left[M(x') \in S\right] + \delta.$$

**Definition 4.2** (Rényi Differential Privacy [Mir17]). *We say $M : \mathcal{X}^* \to \mathcal{Y}$ is $(\alpha, \check{\varepsilon})$-Rényi differentially private $((\alpha, \check{\varepsilon})$-RDP) if, for all $x, x' \in \mathcal{X}^*$ differing only by the addition or removal of one element, we have*

$$\mathrm{D}_\alpha\left(M(x)\|M(x')\right) \leq \check{\varepsilon},$$

*where $\mathrm{D}_\alpha\left(P\|Q\right) := \frac{1}{\alpha-1} \log \mathbb{E}_{Y \leftarrow P}\left[\left(\frac{P(Y)}{Q(Y)}\right)^{\alpha-1}\right]$ denotes the Rényi divergence of order $\alpha$.*

**Definition 4.3** (Concentrated Differential Privacy [DR16; BS16]). *We say $M : \mathcal{X}^* \to \mathcal{Y}$ is $\rho$-zero concentrated differentially private ($\rho$-zCDP) if, for all $x, x' \in \mathcal{X}^*$ differing only by the addition or removal of one element, we have*

$$\forall \alpha > 1 \qquad \mathrm{D}_\alpha\left(M(x)\|M(x')\right) \leq \alpha \cdot \rho.$$

**Remark 4.4.** *In this paper, we focus on to the addition or removal notion of DP, rather than replacement. (In Appendix A, we consider replacement.) Note that, in our theoretical analysis, we consider DP algorithms of the form $M : \{0, 1\}^m \to \mathcal{Y}$. In this case, DP is with respect to flipping one of the input bits, as each bit indicates whether some example is included or excluded.*

The main property of DP that we use is invariance under *postprocessing*. That is, if $M : \mathcal{X}^* \to \mathcal{Y}$ satisfies DP and $F : \mathcal{Y} \to \mathcal{Z}$ is an arbitrary function, then $F \circ M : \mathcal{X}^* \to \mathcal{Z}$ also satisfies DP with the same parameters.

**Gaussian Mechanism** A common method for achieving DP is Gaussian noise addition. The following gives the optimal DP guarantee for the Gaussian mechanism.

**Lemma 4.5** ([BW18, Theorem 8]). *Let $q : \mathcal{X}^* \to \mathbb{R}$ be a function with sensitivity $\Delta := \sup_{x,x'} |q(x) - q(x')|$. (In the supremum $x, x' \in \mathcal{X}^*$ are restricted to differ only by the addition or removal of one element.) Fix $\sigma^2 > 0$ and let $\rho := \Delta^2/2\sigma^2$. Define $M : \mathcal{X}^* \to \mathbb{R}$ by $M(x) = \mathcal{N}(q(x), \sigma^2)$. Then, for any $\varepsilon \geq 0$, the algorithm $M$ satisfies $(\varepsilon, \delta)$-DP with*

$$\delta = \overline{\Phi}\left(\frac{\varepsilon - \rho}{\sqrt{2\rho}}\right) - e^\varepsilon \cdot \overline{\Phi}\left(\frac{\varepsilon + \rho}{\sqrt{2\rho}}\right),$$

*where $\overline{\Phi}(z) := \mathbb{P}_{Z \leftarrow \mathcal{N}(0,1)}[Z > z] = \frac{1}{\sqrt{2\pi}} \int_z^\infty \exp(-x^2/2) \mathrm{d}x$. Furthermore, $M$ satisfies $\rho$-zCDP.*

## 4.2 DP-SGD – Differentially Private Stochastic Gradient Descent

The algorithm whose privacy we are most interested in auditing is Differentially Private Stochastic Gradient Descent (DP-SGD, Algorithm 2). This is the workhorse of private machine learning both in theory [BST14] and in practice [ACGMMTZ16].

---

**Algorithm 2** DP-SGD – Differentially Private Stochastic Gradient Descent

---
1: **Input:** $x \in \mathcal{X}^n$
2: **Model:** Loss function $f : \mathbb{R}^d \times \mathcal{X} \to \mathbb{R}$.
3: **Parameters:** Number of iterations $\ell \geq 1$, clipping threshold $c > 0$, noise multiplier $\sigma > 0$, sampling probability $q \in (0, 1]$, learning rate $\eta > 0$.
4: Initialize $w_0 \in \mathbb{R}^d$.
5: **for** $t = 1, \cdots \ell$ **do**
6:    Sample $S^t \subseteq [n]$ where each $i \in [n]$ is included independently with probability $q$.
7:    Compute $g_i^t = \nabla_{w^{t-1}} f(w^{t-1}, x_i) \in \mathbb{R}^d$ for all $i \in S^t$.
8:    Clip $\hat{g}_i^t = \min\left\{1, \frac{c}{\|g_i^t\|_2}\right\} \cdot g_i^t \in \mathbb{R}^d$ for all $i \in S^t$.
9:    Sample $\xi^t \in \mathbb{R}^d$ from $\mathcal{N}(0, \sigma^2 c^2 I)$.
10:    Sum $\tilde{g}^t = \xi^t + \sum_{i \in S^t} \hat{g}_i^t \in \mathbb{R}^d$.
11:    Update $w^t = w^{t-1} - \eta \cdot \tilde{g}^t \in \mathbb{R}^d$.
12: **end for**
13: **Output:** $w^0, w^1, \cdots, w^\ell$.

---

DP-SGD satisfies differential privacy. Much ink has been spilled precisely quantifying its privacy properties [MTZ19; WBK19; KJH20; GLW21; ZDW22, etc.]. A simple guarantee is the following.

**Proposition 4.6** ([MTZ19; Ste22]). *DP-SGD (Algorithm 2) satisfies $(2, \check{\varepsilon})$-RDP for*

$$\check{\varepsilon} = \ell \cdot \log\left(1 + q^2 \cdot \left(\exp(1/\sigma^2) - 1\right)\right) \approx \ell \cdot q^2 \cdot \frac{1}{\sigma^2}.$$

If $\check{\varepsilon} \leq 1$, then DP-SGD should provide meaningful privacy protection. In particular, $(2, \check{\varepsilon})$-RDP implies that membership inference has a maximum accuracy (in the balanced case) of

$$\frac{1}{2} + \frac{1}{2} \sqrt{\frac{e^{\check{\varepsilon}} - 1}{e^{\check{\varepsilon}} + 3}} \approx \frac{1}{2} + \frac{1}{4} \sqrt{\check{\varepsilon}}. \tag{4}$$

Our goal is to audit this guarantee.

## 4.3 Hypothesis Testing & Statistical Estimation

Our goal is to estimate the privacy parameters of the algorithm that we are auditing. As prior work has noted [DWWZK18; JUO20], this task can be framed as statistical estimation, with a goal of outputting a statistical lower bound on the privacy parameters. These lower bounds will have a corresponding confidence level, roughly representing the probability that the lower bound could have been produced even when analyzing an algorithm with perfect privacy. As empirical methods, it is impossible to have 100% confidence in our methods, so we will generally use 95% confidence in our experiments, comparable to the use of $p < 0.05$ in science literature.

To be precise, our auditor runs the algorithm $M$ and outputs $\varepsilon_{\mathrm{LB}} \geq 0$ with the following guarantee. If $M$ satisfies $(\varepsilon_{\mathrm{true}}, \delta)$-DP, then, with probability at least $1 - \beta$, we have $\varepsilon_{\mathrm{LB}} \leq \varepsilon_{\mathrm{true}}$. Here $1 - \beta$ is the confidence level and $\delta \geq 0$ is fixed. Note that this is a frequentist guarantee, rather than a Bayesian guarantee. That is, the probability is with respect to our auditing procedure, rather than a statement about our beliefs about $M$.

We can also view this in terms of hypothesis testing. Here we start with a "null hypothesis" that $M$ satisfies $(\varepsilon_{\mathrm{null}}, \delta)$-DP and the auditor's goal is to test this hypothesis by running $M$. If the auditor rejects this null hypothesis, then this gives us a lower bound $\varepsilon_{\mathrm{LB}} = \varepsilon_{\mathrm{null}}$.

The difference between hypothesis testing and statistical estimation is that a hypothesis test starts with a given $\varepsilon_{\mathrm{null}}$ and outputs a binary decision to reject or not, while an estimator outputs a number $\varepsilon_{\mathrm{LB}}$. However, we can convert between these:

**Lemma 4.7.** *For each $M$, let $A_M \in \Omega$ be a random variable and let $P_M \in \mathbb{R}$ be a fixed number. For each $\varepsilon, \beta > 0$, let $T_{\varepsilon,\beta} \subset \Omega$ satisfy*

$$\forall M \qquad (P_M = \varepsilon \implies \mathbb{P}[A_M \in T_{\varepsilon,\beta}] \leq \beta). \tag{5}$$

*Further suppose that, if $\varepsilon_1 \leq \varepsilon_2$, then $T_{\varepsilon_1,\beta} \supset T_{\varepsilon_2,\beta}$. Then, for all $M$ and all $\beta > 0$,*

$$\mathbb{P}[P_M \geq \sup \{\varepsilon > 0 : A_M \in T_{\varepsilon,\beta}\}] \geq 1 - \beta. \tag{6}$$

*Proof.* Fix a realization of $A_M$ and suppose $P_M < \sup \{\varepsilon > 0 : A_M \in T_{\varepsilon,\beta}\}$. Then there exists some $\varepsilon \geq P_M$ with $A_M \in T_{\varepsilon,\beta}$ and, hence,

$$A_M \in \bigcup_{\varepsilon \geq P_M} T_{\varepsilon,\beta} = T_{P_M,\beta}.$$

The equality above follows from our monotonicity assumption on $T$. Thus

$$\mathbb{P}\left[P_M < \sup\left\{\varepsilon > 0 : A_M \in T_{\varepsilon,\beta}\right\}\right] \leq \mathbb{P}\left[A_M \in T_{P_M,\beta}\right] \leq \beta,$$

as required. □

To interpret Lemma 4.7, $M$ is an algorithm and $P_M$ is the "true" privacy parameter $\varepsilon$ that it satisfies. (We're considering $\delta$ to be fixed.) The random variable $A_M$ is the output of our auditing procedure applied to $M$. (This is our test statistic in the language of hypothesis testing.) The hypothesis test's rejection set is $T_{\varepsilon,\beta}$ and Equation 5 guarantees that, if $M$ is indeed $(\varepsilon, \delta)$-DP (i.e., the null hypothesis is true), then the probability that we reject the null hypothesis is at most $\beta$. Equation 6 then shows how to estimate the true privacy parameter $P_M$ from $A_M$; we simply take the largest $\varepsilon$ for which we can reject the corresponding null hypothesis.

Note that Lemma 4.7 needs to make a technical monotonicity assumption. In our setting this simply means that, if a given realization of the test statistic $A_M$ allows us to reject the null hypothesis that $M$ is $(\varepsilon_2, \delta)$-DP and $\varepsilon_1 \leq \varepsilon_2$, then we can also reject the null hypothesis that $M$ is $(\varepsilon_1, \delta)$-DP.

## 4.4   Stochastic Dominance

In our theoretical analysis we use the concept of stochastic dominance. Specifically, we use this to formalize the "worst-case" DP algorithm for auditing.

**Definition 4.8** (Stochastic Dominance)**.** *Let $X, Y \in \mathbb{R}$ be random variables. We say $X$ is stochastically dominated by $Y$ (or $Y$ stochastically dominates $X$) if $\mathbb{P}[X > t] \leq \mathbb{P}[Y > t]$ for all $t \in \mathbb{R}$. Equivalently, $X$ is stochastically dominated by $Y$ if there exists a coupling (i.e., a joint distribution that matches the marginal distributions of $X$ and $Y$) such that $\mathbb{P}[X \leq Y] = 1$.*

Stochastic dominance is preserved under sums/convolutions:

**Lemma 4.9.** *Suppose $X_1$ is stochastically dominated by $Y_1$. Suppose that, for all $x \in \mathbb{R}$, the conditional distribution $X_2|X_1 = x$ is stochastically dominated by $Y_2$. Assume that $Y_1$ and $Y_2$ are independent. Then $X_1 + X_2$ is stochastically dominated by $Y_1 + Y_2$.*

*Proof.* For all $t \in \mathbb{R}$, we have

$$
\begin{aligned}
\mathbb{P}[X_1 + X_2 > t] &= \mathop{\mathbb{E}}_{X_1}\left[\mathop{\mathbb{P}}_{X_2}[X_2 > t - X_1 | X_1]\right] \\
&\leq \mathop{\mathbb{E}}_{X_1}\left[\mathop{\mathbb{P}}_{Y_2}[Y_2 > t - X_1]\right] && (Y_2 \text{ dominates } X_2|X_1) \\
&= \mathop{\mathbb{E}}_{Y_2}\left[\mathop{\mathbb{P}}_{X_1}[X_1 > t - Y_2]\right] \\
&\leq \mathop{\mathbb{E}}_{Y_2}\left[\mathop{\mathbb{P}}_{Y_1}[Y_1 > t - Y_2]\right] && (Y_1 \text{ dominates } X_1 \text{ \& independence}) \\
&= \mathbb{P}[Y_1 + Y_2 > t].
\end{aligned}
$$

# 5    Theoretical Analysis

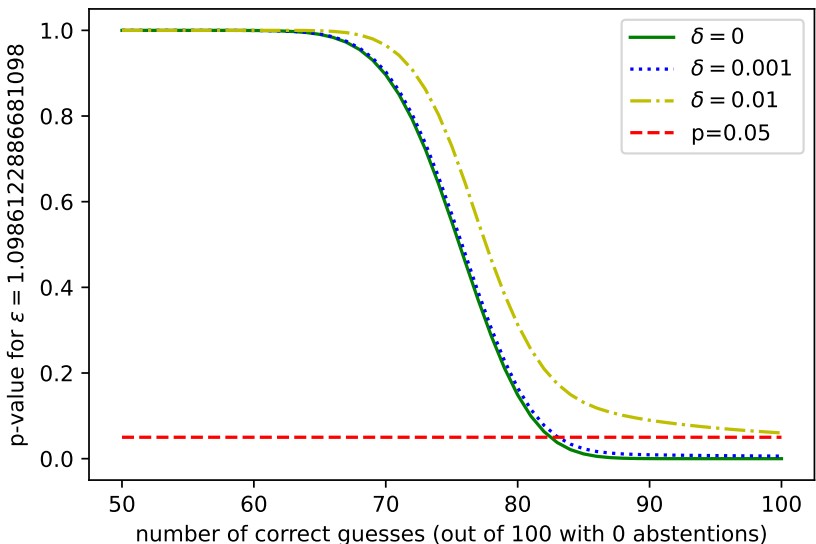

Figure 1: Theorem 5.2's p-value as the number of correct guesses changes for fixed $\varepsilon = \log 3$ (i.e., ideally 75% of guesses correct). The total number of examples and guesses is 100.

To analyze the results of our audit, we leverage the connection between DP and generalization [DFHPRR15b; DFHPRR15a; BNSSSU16; RRST16; JLNRSMS19; SZ20]. Unfortunately, directly applying the existing results from the literature is unlikely to yield meaningful results, as the constants are not optimal. Thus we provide an analysis of DP's generalization guarantees that is suitable for our application and which has sharp constants.

We consider the following formalism. The algorithm $M : \{-1, +1\}^m \to \mathbb{R}^m$ takes in a vector of bits and outputs a vector of "guesses". Each input bit indicates whether or not a particular example is included in or excluded from the dataset. In particular, the DP guarantee ensures that the outputs are indistinguishable if we flip one bit, which corresponds to adding or removing the corresponding data point. Each coordinate of the output is a guess for the corresponding input bit; the sign of the score should match the corresponding input bit, while the magnitude is a reflection of the confidence.

The algorithm $M$ represents both the "real" algorithm (e.g., DP-SGD) and the auditor which postprocesses the output of the real algorithm into guesses. In this formalism, the examples themselves are considered fixed and not part of the input – i.e., the examples are "hardcoded" into $M$. The algorithm $M$ is an abstraction for our analysis, rather than a realistic system.

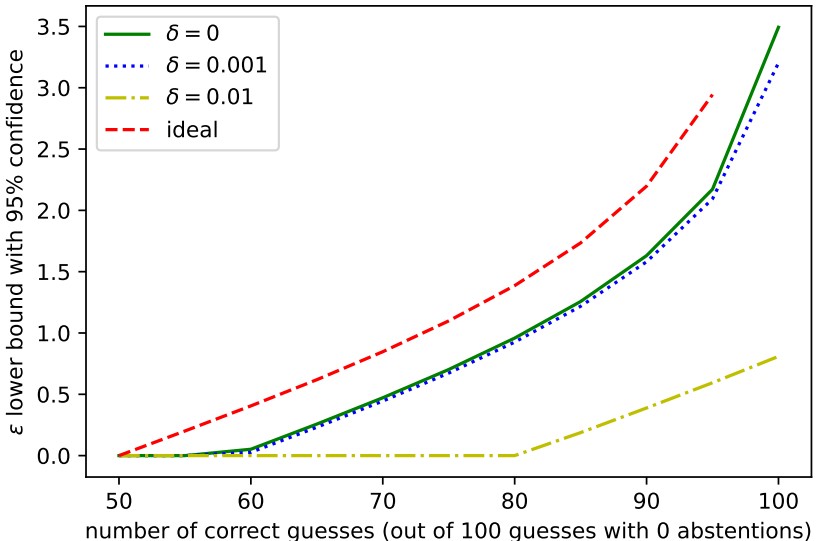

Figure 2: Lower bound on the privacy parameter $\varepsilon$ given by Theorem 5.2 with 95% confidence as the number of correct guesses changes. The total number of examples and guesses is 100. For comparison, we plot the `ideal` $\varepsilon$ that gives $100 \cdot \frac{e^\varepsilon}{e^\varepsilon+1}$ correct guesses.

We evaluate the quantity

$$W := \sum_i^m \max\{0, T_i \cdot S_i\},$$

where $S$ is uniform on $\{-1, +1\}^m$ and $T = M(S)$. If $T_i$ and $S_i$ disagree in sign (i.e., the guess is wrong), then $\max\{0, T_i \cdot S_i\} = 0$; if they agree (i.e., the guess is right), then $\max\{0, T_i \cdot S_i\} = |T_i|$. That is, $W$ increases when we guess correctly and the increase is proportional to how much "weight" we placed on that guess. The auditor seeks to maximize $W$ and then we compare it to a baseline that is consistent with DP. (The analysis in this section focuses on computing this baseline.) Incorrect guesses do not increase $W$, but they do increase the baseline. Note that we can guess $T_i = 0$, which amounts to abstaining from making a guess; this doesn't increase $W$, but also doesn't increase the baseline.

Our formalism is inspired by that of Steinke and Zakynthinou [SZ20], who also restrict to binary inputs. In contrast, most of the work connecting DP and generalization does not do this. The benefit of restricting to binary inputs which represent inclusion or exclusion of a data point is that it simplifies our analysis.

## 5.1 Pure DP Analysis

We first consider the pure DP ($\delta = 0$) case, as it is considerably simpler than the general case. We follow the analysis of Jung, Ligett, Neel, Roth, Sharifi-Malvajerdi, and Shenfeld

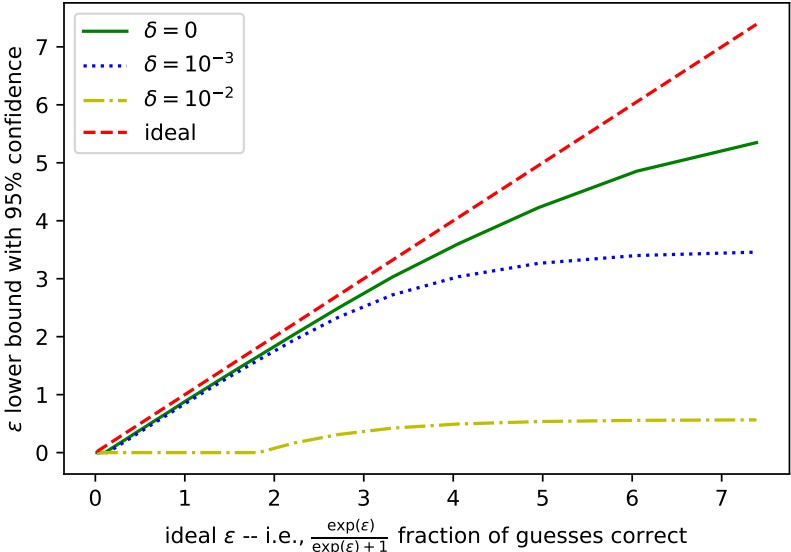

Figure 3: Lower bound on the privacy parameter $\varepsilon$ given by Theorem 5.2 with 95% confidence as the number of correct guesses changes. The total number of examples and guesses is 1000 (with no abstentions). Here we plot the ideal $\varepsilon$ on the horizontal axis, so that the number of correct guesses is $1000 \cdot \frac{e^\varepsilon}{e^\varepsilon + 1}$.

[JLNRSMS19] with some refinement. Specifically, rather than relying on a Hoeffding bound, we show that it is stochastically dominated by a Binomial distribution. This result is tight – i.e., if $M$ independently performs a randomized response for each input bit, then the inequality becomes an equality.

**Proposition 5.1** (Pure DP Version of Main Result). *Let $M : \{-1, +1\}^m \to \mathbb{R}^m$ satisfy $(\varepsilon, 0)$-DP. Let $S \in \{-1, +1\}^m$ be uniformly random. Let $T = M(S) \in \mathbb{R}^m$. Then, for all $v \in \mathbb{R}$ and all $t \in \mathbb{R}^m$ in the support of $T$,[3]*

$$\mathop{\mathbb{P}}_{\substack{S \leftarrow \{-1,1\}^m \\ T \leftarrow M(S)}} \left[ \sum_i^m \max\{0, T_i \cdot S_i\} \geq v \,\middle|\, T = t \right] \leq \mathop{\mathbb{P}}_{\check{S} \leftarrow \mathsf{Bernoulli}\left(\frac{e^\varepsilon}{e^\varepsilon + 1}\right)^m} \left[ \sum_i^m \check{S}_i \cdot |t_i| \geq v \right] =: \beta(m, \varepsilon, v, t).$$

Proposition 5.1 is Bayesian: We condition on the output and then consider the probability that each guess was right. The vector $\check{S}$ should be seen as indicating whether each guess was right. The proposition says that, in the worst case, each guess is correct independently with probability $\frac{e^\varepsilon}{e^\varepsilon + 1}$.

---

[3]To be precise, this holds with probability 1, but may fail for $t$ in a set of measure zero under $T$. Note that $S \leftarrow \{-1, 1\}^m$ denotes that $S$ is uniform on the set $\{-1, 1\}^m$ and $\check{S} \leftarrow \mathsf{Bernoulli}\left(\frac{e^\varepsilon}{e^\varepsilon + 1}\right)$ denotes that $\check{S} \in \{0, 1\}^m$ has a product distribution with each coordinate having expectation $\frac{e^\varepsilon}{e^\varepsilon + 1}$.

How do we use this result? Suppose we have conducted an audit and observed $s$ and $t$ as the output of Algorithm 3. Let $v = \sum_i^m \max\{0, s_i \cdot t_i\}$. Following Lemma 4.7, we choose a desired confidence $1 - \beta < 1$ (e.g., $\beta = 0.05$) and then we choose $\varepsilon \geq 0$ so that $\beta(m, \varepsilon, v, t) = \beta$. Then this value of $\varepsilon$ is our lower bound.

In the language of hypothesis testing, $W = \sum_i^m \max\{0, T_i \cdot S_i\}$ is the test statistic and our null hypothesis is that $M$ is $\varepsilon$-DP. Under the null hypothesis we have $\mathbb{P}[W \geq v] \leq \beta(m, \varepsilon, v, t)$. Thus, if $v$ is the observed value of the test statistic, then $\beta(m, \varepsilon, v, t)$ is our p-value. And we can reject the null hypothesis if, say, $\beta(m, \varepsilon, v) \leq 0.05$.

*Proof.* Fix some $t \in \mathbb{R}^m$. We now analyze the distribution of $S$ conditioned on $M(S) = t$. Note that the unconditional distribution of $S$ is uniform on $\{-1, +1\}^m$ and $M$ is $(\varepsilon, 0)$-DP. We perform the analysis one bit at a time. Fix some $i \in [m]$ and $s_{<i} \in \{-1, +1\}^{i-1}$. By Bayes' law and $(\varepsilon, 0)$-DP,

$$
\begin{aligned}
&\mathbb{P}\left[S_i = 1 | M(S) = t, S_{<i} = s_{<i}\right] \\
&= \frac{\mathbb{P}\left[M(S) = t | S_i = 1, S_{<i} = s_{<i}\right] \cdot \mathbb{P}\left[S_i = 1 | S_{<i} = s_{<i}\right]}{\mathbb{P}\left[M(S) = t | S_{<i} = s_{<i}\right]} \\
&= \frac{\mathbb{P}\left[M(S) = t | S_i = 1, S_{<i} = s_{<i}\right] \cdot \mathbb{P}\left[S_i = 1\right]}{\mathbb{P}\left[M(S) = t | S_i = 1, S_{<i} = s_{<i}\right] \cdot \mathbb{P}\left[S_i = 1\right] + \mathbb{P}\left[M(S) = t | S_i = -1, S_{<i} = s_{<i}\right] \cdot \mathbb{P}\left[S_i = -1\right]} \\
&= \frac{\mathbb{P}\left[M(S) = t | S_i = 1, S_{<i} = s_{<i}\right] \cdot \frac{1}{2}}{\mathbb{P}\left[M(S) = t | S_i = 1, S_{<i} = s_{<i}\right] \cdot \frac{1}{2} + \mathbb{P}\left[M(S) = t | S_i = -1, S_{<i} = s_{<i}\right] \cdot \frac{1}{2}} \\
&= \frac{1}{1 + \mathbb{P}\left[M(S) = t | S_i = -1, S_{<i} = s_{<i}\right] / \mathbb{P}\left[M(S) = t | S_i = 1, S_{<i} = s_{<i}\right]} \\
&\in \left[\frac{1}{1 + e^\varepsilon}, \frac{1}{1 + e^{-\varepsilon}}\right].
\end{aligned}
$$

Thus $\mathbb{P}\left[S_i = \mathsf{sign}(T_i) | T = t, S_{<i} = s_{<i}\right] \leq \frac{1}{1 + e^{-\varepsilon}} = \frac{e^\varepsilon}{e^\varepsilon + 1}$.

With this in hand, we can prove the result by induction. We assume inductively that $W_{m-1} := \sum_i^{m-1} \max\{0, T_i \cdot S_i\}$ is stochatiscally dominated by $\check{W}_{m-1} := \sum_i^{m-1} \check{S}_i \cdot |t_i|$ where $\check{S} \leftarrow \mathsf{Bernoulli}\left(\frac{e^\varepsilon}{e^\varepsilon + 1}\right)^{m-1}$. As above, conditioned on the value of $W_{m-1}$, the variable $\max\{0, T_m \cdot S_m\} = |T_m| \cdot \mathbb{I}[S_m = \mathsf{sign}(T_m)]$ is stochastically dominated by $|T_m| \cdot \mathsf{Bernoulli}\left(\frac{e^\varepsilon}{e^\varepsilon + 1}\right)$. By Lemma 4.9, $W_m = W_{m-1} + \max\{0, T_m \cdot S_m\}$ is stochastically dominated by $\check{W}_m := \sum_i^m \check{S}_i \cdot |t_i|$ where $\check{S} \leftarrow \mathsf{Bernoulli}\left(\frac{e^\varepsilon}{e^\varepsilon + 1}\right)^m$. $\qquad\square$

## 5.2 Approximate DP Analysis

We extend the pure DP analysis (§5.1) to approximate DP ($\delta > 0$). This becomes quite messy. In the pure DP case, we can condition on an arbitrary output $t$. In the approximate DP case, some outputs are "bad" in the sense that the privacy loss is unbounded. To handle this we do two things: First, we require the guesses to be bounded (i.e., $T \in [-1, +1]^m$ instead of $T \in \mathbb{R}^m$), which ensures that bad outputs cannot skew things too much. Second, the guarantees we prove have an additional failure probability that depends on $\delta$.

Our analysis most closely resembles that of Rogers, Roth, Smith, and Thakkar [RRST16]. Essentially, we repeat the analysis for the pure DP case, but add some failure events, and carefully account for how much they can distort the results.

**Theorem 5.2** (Main Result). *Let $M : \{-1,+1\}^m \to [-1,+1]^m$ satisfy $(\varepsilon,\delta)$-DP. Let $S \in \{-1,+1\}^m$ be uniformly random. Let $T = M(S) \in [-1,+1]^m$. Then, for all $v \in \mathbb{R}$,*

$$\mathbb{P}_{\substack{S \leftarrow \{-1,+1\}^m \\ T \leftarrow M(S)}} \left[ \sum_i^m \max\{0, T_i \cdot S_i\} \geq v \right] \leq \beta + \alpha \cdot 2m \cdot \delta, \tag{7}$$

*where*

$$\beta = \mathbb{P}_{\check{W}^*} \left[ \check{W}^* \geq v \right], \tag{8}$$

$$\alpha = \max \left\{ \frac{1}{i} \left( \mathbb{P}_{\check{W}^*} \left[ \check{W}^* \geq v - i \right] - \beta \right) : i \in \{1, 2, \cdots, m\} \right\}. \tag{9}$$

*Here $\check{W}^*$ is any distribution on $\mathbb{R}$ that stochastically dominates $\check{W}(t) := \sum_i^m \check{S}_i |t_i|$ for $\check{S} \leftarrow$ Bernoulli $\left( \frac{e^\varepsilon}{e^\varepsilon+1} \right)^m$ for all $t$ in the support of $T$.*

To evaluate the bound of Theorem 5.2, we need to identify $\check{W}^*$ and compute its distribution. We can set $\mathbb{P}_{\check{W}^*} \left[ \check{W}^* \geq v \right] = \sup_{t \in \mathsf{support}(T)} \mathbb{P}_{\check{W}} \left[ \check{W}(t) \geq v \right]$. This can be difficult to compute, depending on what we know about the support of $T$. If the support of $T$ is nice, we can compute this explicitly; e.g., see Corollary 5.4. There are other things we can do. For example, if we have bounds on $\sup_{t \in \mathsf{support}(T)} \|t\|_2$ and $\sup_{t \in \mathsf{support}(T)} \|t\|_1$, then we can use a concentration inequality to bound $\sup_{t \in \mathsf{support}(T)} \mathbb{P}_{\check{W}} \left[ \check{W}(t) \geq v \right]$ and then use this bound as the distribution of $\check{W}^*$. This yields the following corollary.

**Corollary 5.3** (Analytic Version of Main Result). *Let $M : \{-1,+1\}^m \to [-1,+1]^m$ satisfy $(\varepsilon,\delta)$-DP. Let $S \in \{-1,+1\}^m$ be uniformly random. Let $T = M(S) \in [-1,+1]^m$. Suppose $\mathbb{P}[\|T\|_2 \leq r_2] = 1$ and $\mathbb{P}[\|T\|_1 \leq r_1] = 1$. Then, for all $v \geq \frac{e^\varepsilon}{e^\varepsilon+1} \cdot r_1 + 2$, we have*

$$\mathbb{P}_{\substack{S \leftarrow \{-1,+1\}^m \\ T \leftarrow M(S)}} \left[ \sum_i^m \max\{0, T_i \cdot S_i\} \geq v \right] \leq f(v) + 2m \cdot \delta \cdot \max \left\{ \frac{f(v-i) - f(v)}{i} : i \in [m] \right\} \tag{10}$$

$$\leq f(v) + 2m\delta \cdot \max \left\{ \frac{2}{v - \frac{e^\varepsilon}{e^\varepsilon+1} r_1}, f\left( \frac{1}{2} \left( v + \frac{e^\varepsilon}{e^\varepsilon+1} r_1 \right) \right) \right\}, \tag{11}$$

*where*

$$f(v) := \left\{ \begin{array}{ll} \exp\left( \frac{-2}{r_2^2} \left( v - \frac{e^\varepsilon}{e^\varepsilon+1} r_1 \right)^2 \right) & \text{if } v \geq \frac{e^\varepsilon}{e^\varepsilon+1} r_1 \\ 1 & \text{if } v < \frac{e^\varepsilon}{e^\varepsilon+1} r_1 \end{array} \right\}.$$

In particular, if we substitute $v = \frac{e^\varepsilon}{e^\varepsilon+1}r_1 + r_2 \cdot \sqrt{\frac{1}{2}\log(1/\beta)}$ into Equation 11, we get

$$\mathop{\mathbb{P}}_{\substack{S\leftarrow\{-1,+1\}^m \\ T\leftarrow M(S)}} \left[\sum_i^m \max\{0, T_i \cdot S_i\} \geq v\right] \leq \beta + 2m \cdot \delta \cdot \max\left\{\frac{1}{r_2\sqrt{\frac{1}{2}\log(1/\beta)}}, \beta^{1/4}\right\}. \quad (12)$$

*Proof.* Fix an arbitrary $t$ the support of $T$. Define $\check{W}(t) := \sum_i^m \check{S}_i|t_i|$ for $\check{S} \leftarrow \mathsf{Bernoulli}\left(\frac{e^\varepsilon}{e^\varepsilon+1}\right)^m$. Then $\mathbb{E}\left[\check{W}(t)\right] = \frac{e^\varepsilon}{e^\varepsilon+1}\|t\|_1$. By Hoeffding's inequality, for all $\lambda \geq 0$,

$$\mathbb{P}\left[\check{W}(t) \geq \frac{e^\varepsilon}{e^\varepsilon+1}\|t\|_1 + \lambda\right] \leq \exp\left(\frac{-2\lambda^2}{\|t\|_2^2}\right).$$

Now define $\check{W}^*$ by

$$\mathbb{P}\left[\check{W}^* \geq v\right] := f(v) := \left\{\begin{array}{ll} \exp\left(\frac{-2}{r_2^2}\left(v - \frac{e^\varepsilon}{e^\varepsilon+1}r_1\right)^2\right) & \text{if } v \geq \frac{e^\varepsilon}{e^\varepsilon+1}r_1 \\ 1 & \text{if } v < \frac{e^\varepsilon}{e^\varepsilon+1}r_1 \end{array}\right\}.$$

Since $\check{W}^*$ stochastically dominates $\check{W}(t)$ for all $t$ in the support of $T$, we can apply Theorem 5.2 to obtain the first part of the result (10).

Next, for any $c \geq 1$, we have

$$\max\left\{\frac{f(v-i) - f(v)}{i} : i \in [m]\right\}$$

$$\leq \max\left\{\frac{f(v-x)}{x} : x \in [1, \infty)\right\} \qquad\qquad (f(v) \geq 0 \text{ and } [m] \subset [1, \infty))$$

$$= \max\left\{\max\left\{\frac{f(v-x)}{x} : x \in [1, c]\right\}, \max\left\{\frac{f(v-x)}{x} : x \in [c, \infty)\right\}\right\}$$

$$\leq \max\left\{\max\left\{\frac{f(v-x)}{1} : x \in [1, c]\right\}, \max\left\{\frac{1}{x} : x \in [c, \infty)\right\}\right\}$$

$$= \max\left\{f(v-c), \frac{1}{c}\right\}.$$

Setting $c = \frac{1}{2}\left(v - \frac{e^\varepsilon}{e^\varepsilon+1}r_1\right)$ yields the second part of the result (11) $\qquad\square$

In the next corollary we restrict $M$ to ternary outputs, so it must either guess ($T_i = \pm 1$) or abstain ($T_i = 0$). We bound the number of guesses by $r$. In this case the dominating distribution $\check{W}^*$ is a binomial distribution, which is relatively easy to compute. This is the form of Theorem 5.2 that we use in all of our experimental results. We provide pseudocode in Appendix D.

**Corollary 5.4** (Ternary Guesses). *Let $M : \{-1, +1\}^m \to \{-1, 0, +1\}^m$ satisfy $(\varepsilon, \delta)$-DP. Let $S \in \{-1, +1\}^m$ be uniformly random. Let $T = M(S) \in \{-1, 0, +1\}^m$. Suppose $\mathbb{P}\left[\|T\|_1 \leq r\right] = 1$. Then, for all $v \in \mathbb{R}$,*

$$\mathop{\mathbb{P}}_{\substack{S\leftarrow\{-1,+1\}^m \\ T\leftarrow M(S)}} \left[\sum_i^m \max\{0, T_i \cdot S_i\} \geq v\right] \leq f(v) + 2m \cdot \delta \cdot \max\left\{\frac{f(v-i) - f(v)}{i} : i \in \{1, 2, \cdots, m\}\right\},$$

*where*

$$f(v) := \mathbb{P}_{\check{W} \leftarrow \mathsf{Binomial}\left(r, \frac{e^\varepsilon}{e^\varepsilon+1}\right)} \left[\check{W} \geq v\right].$$

Now we delve into the proof of Theorem 5.2. We use a decomposition result of Kairouz, Oh, and Viswanath [KOV15] (see also [MV15] & [Ste22, Corollary 24]).

**Lemma 5.5.** *Let $P$ and $Q$ be probability distributions over $\mathcal{Y}$. Fix $\varepsilon, \delta \geq 0$. Suppose that, for all measurable $S \subset \mathcal{Y}$, we have $P(S) \leq e^\varepsilon \cdot Q(S) + \delta$ and $Q(S) \leq e^\varepsilon P(S) + \delta$.*

*Then there exist $\delta' \in [0, \delta]$ and distributions $P'$, $Q'$, $P''$, and $Q''$ over $\mathcal{Y}$ such that the following three properties are all satisfied. First, we can express $P$ and $Q$ as convex combinations:*

$$P = (1 - \delta')P' + \delta'P'',$$
$$Q = (1 - \delta')Q' + \delta'Q''.$$

*Second, for all measurable $S \subset \mathcal{Y}$, we have $e^{-\varepsilon}P'(S) \leq Q'(S) \leq e^\varepsilon P'(S)$. Third, there exist measurable $S, T \subset \mathcal{Y}$ such that $P''(S) = 1$, $Q''(T) = 1$, $\forall S' \subset S \; P(S') \geq Q(S')$, and $\forall T' \subset T \; Q(T') \geq P(T')$.*

*Proof.* This proof follows that of Steinke [Ste22]. We begin with some formalities: Fix some base measure such that $P$ and $Q$ are absolutely continuous with respect to the base measure. (If $P$ and $Q$ are discrete distributions, this can be the counting measure. If they are continuous distributions, this can be the Lebesgue measure. In general, $P + Q$ serves as such a measure.) For $y \in \mathcal{Y}$, let $P(y)$ and $Q(y)$ denote the Radon-Nikodym derivative of $P$ and, respectively, $Q$ with respect to this base measure.

If $e^{-\varepsilon} \cdot Q(S) \leq P(S) \leq e^\varepsilon \cdot Q(S)$ for all measurable $S$, then the result follows trivially by setting $\delta' = 0$, $P' = P$ and $Q' = Q$, and choosing $P''$ and $Q''$ to be arbitrary distributions supported on $S = \{y \in \mathcal{Y} : P(y) \geq Q(y)\}$ and $T = \{y \in \mathcal{Y} : P(y) \leq Q(y)\}$ respectively. Thus we assume that this is not the case and, hence, that $\delta > 0$ and $d_{\mathrm{TV}}(P, Q) > 0$.

Similarly, if $\delta \geq 1$ and $d_{\mathrm{TV}}(P, Q) = 1$, then the result follows trivially by setting $\delta' = 1$, $P'' = P$, $Q'' = Q$, and $P' = Q'$ arbitrary. Thus we assume that $\min\{\delta, d_{\mathrm{TV}}(P, Q)\} < 1$.

Fix $\varepsilon_1, \varepsilon_2 \in [0, \varepsilon]$ to be determined later. Define distributions $P'$, $P''$, $Q'$, and $Q''$ (in terms of their Radon-Nikodym derivatives) as follows. For all points $y \in \mathcal{Y}$,

$$P'(y) = \frac{\min\{P(y), e^{\varepsilon_1} \cdot Q(y)\}}{1 - \delta_1},$$
$$P''(y) = \frac{P(y) - (1 - \delta_1)P'(y)}{\delta_1} = \frac{\max\{0, P(y) - e^{\varepsilon_1} \cdot Q(y)\}}{\delta_1},$$
$$Q'(y) = \frac{\min\{Q(y), e^{\varepsilon_2} \cdot P(y)\}}{1 - \delta_2},$$
$$Q''(y) = \frac{Q(y) - (1 - \delta_2)Q'(y)}{\delta_2} = \frac{\max\{0, Q(y) - e^{\varepsilon_2} \cdot P(y)\}}{\delta_2},$$

where $\delta_1, \delta_2 \in (0, 1)$ are appropriate normalizing constants. (We will choose $\varepsilon_1$ to avoid $\delta_1 \in \{0, 1\}$ and, likewise, we will choose $\varepsilon_2$ to avoid $\delta_2 \in \{0, 1\}$.)

By construction, $(1 - \delta_1)P' + \delta_1 P'' = P$ and $(1 - \delta_2)Q' + \delta_2 Q'' = Q$, so the first property is satisfied. Note that $P''$ is supported on $S = \{y \in \mathcal{Y} : P(y) > e^{\varepsilon_1} \cdot Q(y)\}$ and $Q''$ is supported on $T = \{y \in \mathcal{Y} : Q(y) > e^{\varepsilon_2} \cdot P(y)\}$, which implies the third property.

If $0 < \delta_1 = \delta_2 \le \delta$, then we have the appropriate decomposition (with $\delta' = \delta_1 = \delta_2$) and, for all $y \in \mathcal{Y}$, we have

$$e^{-\varepsilon} \le e^{-\varepsilon_2} \le \frac{P'(y)}{Q'(y)} = \frac{\min\{P(y), e^{\varepsilon_1} \cdot Q(y)\}}{\min\{Q(y), e^{\varepsilon_2} \cdot P(y)\}} \le e^{\varepsilon_1} \le e^{\varepsilon},$$

as required for the second property.

It only remains to show that we can ensure that $0 < \delta_1 = \delta_2 \le \delta$ by appropriately setting $\varepsilon_1, \varepsilon_2 \in [0, \varepsilon]$. We have

$$\delta_1 = \int_{\mathcal{Y}} \max\{0, P(y) - e^{\varepsilon_1} \cdot Q(y)\} \mathrm{d}y = \int_S P(y) - e^{\varepsilon_1} \cdot Q(y) \mathrm{d}y = P(S) - e^{\varepsilon_1} Q(S),$$

where $S = \{y \in \mathcal{Y} : P(y) \ge e^{\varepsilon_1} \cdot Q(y)\}$. If $\varepsilon_1 = \varepsilon$, then $\delta_1 \le \delta$ by assumption. If $\varepsilon_1 = 0$, then $\delta_1 = d_{\mathrm{TV}}(P, Q) > 0$. By decreasing $\varepsilon_1$, we continuously increase $\delta_1$. Thus, by starting at $\varepsilon_1 = \varepsilon$ and decreasing $\varepsilon_1$ until either $\varepsilon_1 = 0$ or $\delta_1 = \delta$, we can pick $\varepsilon_1 \in [0, \varepsilon]$ such that $\delta_1 = \min\{\delta, d_{\mathrm{TV}}(P, Q)\} \in (0, 1)$. Similarly, we can pick $\varepsilon_2 \in [0, \varepsilon]$, such that $\delta_2 = \min\{\delta, d_{\mathrm{TV}}(P, Q)\}$. $\qquad\square$

We need a Bayesian version of this decomposition. I.e., suppose we observe a sample from either $P$ or $Q$ and we have a prior on these two possibilities, what is the posterior distribution on possibilities? The following gives such a result. However, it introduces an event $E_{P,Q}$. Intuitively, when $E_{P,Q}(Y) = 1$, then we get the result we would get under pure DP. But $E_{P,Q}(Y) = 0$ with probability $\delta$, in which case things can fail arbitrarily.

Kasiviswanathan and Smith [KS14, Lemma 3.4] provide a similar result. Ours improves the constant factors and is also stated slightly differently.

**Lemma 5.6.** *Let $P$ and $Q$ be probability distributions over $\mathcal{Y}$. Fix $\varepsilon, \delta \ge 0$. Suppose that, for all measurable $S \subset \mathcal{Y}$, we have $P(S) \le e^{\varepsilon} \cdot Q(S) + \delta$ and $Q(S) \le e^{\varepsilon} P(S) + \delta$.*

*Then there exists a randomized function $E_{P,Q} : \mathcal{Y} \to \{0, 1\}$ with the following properties.*

*Fix $p \in [0, 1]$ and suppose $X \leftarrow \mathsf{Bernoulli}(p)$. If $X = 1$, sample $Y \leftarrow P$; and, if $X = 0$, sample $Y \leftarrow Q$. Then, for all $y \in \mathcal{Y}$, we have*

$$\mathbb{P}_{\substack{X \leftarrow \mathsf{Bernoulli}(p) \\ Y \leftarrow XP + (1-X)Q}} [X = 1 \wedge E_{P,Q}(Y) = 1 | Y = y] \le \frac{p}{p + (1 - p)e^{-\varepsilon}}.$$

*Furthermore,*

$$\mathbb{E}_{Y \leftarrow P} [E_{P,Q}(Y)] \ge 1 - \delta \qquad \text{and} \qquad \mathbb{E}_{Y \leftarrow Q} [E_{P,Q}(Y)] \ge 1 - \delta.$$

*Proof.* We apply the decomposition from Lemma 5.5: There exist distributions $P'$, $Q'$, $P''$, and $Q''$ over $\mathcal{Y}$ and $\delta' \in [0, \delta]$ such that

$$P = (1 - \delta')P' + \delta'P'',$$
$$Q = (1 - \delta')Q' + \delta'Q'',$$

and, for all $y \in \mathcal{Y}$, $e^{-\varepsilon}P'(y) \leq Q'(y) \leq e^{\varepsilon}P'(y)$ and $P''(y) > 0 \implies P(y) \geq Q(y)$ and $Q''(y) > 0 \implies P(y) \leq Q(y)$. (Here $P(\cdot)$ denotes the Radon-Nikodym derivative of the distribution $P$ with respect to some appropriate base measure and similarly for the other distributions.)

We define $E_{P,Q} : \mathcal{Y} \to \{0, 1\}$ by

$$\mathbb{P}\left[E_{P,Q}(y) = 1\right] = (1 - \delta') \cdot \frac{P'(y)}{P(y)} = 1 - \delta' \cdot \frac{P''(y)}{P(y)}.$$

Clearly, $\underset{Y \leftarrow P, E_{P,Q}}{\mathbb{E}}[E_{P,Q}(Y)] = \int_{\mathcal{Y}} P(y)\mathbb{P}\left[E_{P,Q}(Y) = 1\right]\mathrm{d}y = \int_{\mathcal{Y}}(1 - \delta')P'(y)\mathrm{d}y = 1 - \delta' \geq 1 - \delta$.
Also

$$\begin{aligned}
\underset{Y \leftarrow Q}{\mathbb{E}}[E_{P,Q}(Y)] &= 1 - \delta' \underset{Y \leftarrow Q}{\mathbb{E}}\left[\frac{P''(y)}{P(y)}\right] \\
&= 1 - \delta' \int_{\mathcal{Y}} \frac{Q(y)}{P(y)} \cdot P''(y)\mathrm{d}y \\
&\geq 1 - \delta' \int_{\mathcal{Y}} P''(y)\mathrm{d}y \\
&= 1 - \delta' \geq 1 - \delta,
\end{aligned}$$

since $P''(y) > 0 \implies P(y) \geq Q(y)$. For any $y \in \mathcal{Y}$, we have

$$\mathbb{P}\left[X = 1 \wedge E_{P,Q}(Y) = 1 | Y = y\right]$$
$$= \mathbb{P}\left[X = 1 | Y = y\right] \cdot \mathbb{P}\left[E_{P,Q}(y) = 1\right]$$
$$= \frac{\mathbb{P}\left[Y = y | X = 1\right] \cdot \mathbb{P}\left[X = 1\right]}{\mathbb{P}\left[Y = y\right]} \cdot \mathbb{P}\left[E_{P,Q}(y) = 1\right]$$
$$= \frac{P(y) \cdot p}{pP(y) + (1-p)Q(y)} \cdot \mathbb{P}\left[E_{P,Q}(y) = 1\right]$$
$$= \frac{p(1-\delta')P'(y) + p\delta'P''(y)}{p(1-\delta')P'(y) + p\delta'P''(y) + (1-p)(1-\delta')Q'(y) + (1-p)\delta'Q''(y)} \cdot \mathbb{P}\left[E_{P,Q}(y) = 1\right]$$
$$= \frac{p + p\frac{\delta'P''(y)}{(1-\delta')P'(y)}}{p + p\frac{\delta'P''(y)}{(1-\delta')P'(y)} + (1-p)\frac{Q'(y)}{P'(y)} + (1-p)\frac{\delta'Q''(y)}{(1-\delta')P'(y)}} \cdot \mathbb{P}\left[E_{P,Q}(y) = 1\right]$$
$$= \frac{p + p\frac{\delta'}{1-\delta'}\frac{P''(y)}{P'(y)}}{p + (1-p)\frac{Q'(y)}{P'(y)} + \frac{\delta'}{1-\delta'} \cdot \frac{pP''(y) + (1-p)Q''(y)}{P'(y)}} \cdot \mathbb{P}\left[E_{P,Q}(y) = 1\right]$$
$$\leq \frac{p + p\frac{\delta'}{1-\delta'}\frac{P''(y)}{P'(y)}}{p + (1-p)e^{-\varepsilon} + 0} \cdot \mathbb{P}\left[E_{P,Q}(y) = 1\right]$$
$$= \frac{p}{p + (1-p)e^{-\varepsilon}} \cdot \left(1 + \frac{\delta'}{1-\delta'}\frac{P''(y)}{P'(y)}\right) \cdot \mathbb{P}\left[E_{P,Q}(y) = 1\right]$$
$$= \frac{p}{p + (1-p)e^{-\varepsilon}} \cdot \left(\frac{P(y)}{(1-\delta')P'(y)}\right) \cdot \mathbb{P}\left[E_{P,Q}(y) = 1\right]$$
$$= \frac{p}{p + (1-p)e^{-\varepsilon}}.$$

$\square$

Now we can prove an analog of Proposition 5.1 for the $(\varepsilon, \delta)$-DP setting.

**Proposition 5.7** (General Form of Main Result). *Let $M : \{-1, +1\}^m \to [-1, +1]^m$ sat-isfy $(\varepsilon, \delta)$-DP. Let $S \in \{-1, +1\}^m$ be $m$ independent samples from $2\mathsf{Bernoulli}(p) - 1$ – i.e., $\mathbb{P}\left[S_i = 1\right] = p$ independently for each $i \in [m]$. Let $T = M(S) \in [-1, +1]^m$. Then, for all $v \in \mathbb{R}$ and all $t \in [-1, +1]^m$,*

$$\mathbb{P}_{\substack{S \leftarrow (2\mathsf{Bernoulli}(p)-1)^m, \\ T \leftarrow M(S)}}\left[\sum_i^m \max\{0, T_i \cdot S_i\} \geq v \Big| T = t\right] \leq$$

$$\mathbb{P}_{\substack{\check{S}^+ \leftarrow \mathsf{Bernoulli}\left(\frac{p \cdot e^\varepsilon}{p \cdot e^\varepsilon + 1 - p}\right)^m, \\ \check{S}^- \leftarrow \mathsf{Bernoulli}\left(\frac{(1-p) \cdot e^\varepsilon}{(1-p) \cdot e^\varepsilon + p}\right)^m, F}}\left[F(t) + \sum_{i \in [m]: t_i > 0} t_i \cdot \check{S}_i^+ + \sum_{i \in [m]: t_i < 0} -t_i \cdot \check{S}_i^- \geq v\right],$$

*where $F : [-1, 1]^m \to \{0, 1, \cdots, m\}$ is independent from $\check{S}^+$ and $\check{S}^-$ and satisfies $\mathbb{E}_{T,F}\left[F(T)\right] \leq 2m \cdot \delta$.*

*Proof.* For $i \in [m] \cup \{0\}$ and $s_{\leq i} \in \{-1, 1\}^i$, let $M(s_{\leq i})$ denote the distribution on $[-1, 1]^m$ obtained by conditioning $M(S)$ on $S_{\leq i} = s_{\leq i}$. We can express this as a convex combination:

$$M(s_{\leq i}) = \sum_{s_{>i} \in \{-1,1\}^{m-i}} M(s_{\leq i}, s_{>i}) \cdot \underset{S_{>i} \leftarrow (2\text{Bernoulli}(p)-1)^{m-i}}{\mathbb{P}} [S_{>i} = s_{>i}].$$

For distributions $P$ and $Q$ on $[-1, 1]^m$, let $E_{P,Q} : [-1, 1]^m \to \{0, 1\}$ be the randomized function promised by Lemma 5.6. In our analysis, the internal randomness of $E_{P,Q}$ is independent from everything else – i.e., the only dependence is induced by its input. Specifically, for all $i \in [m]$, all $s_{<i} \in \{-1, 1\}^{i-1}$, and all $t \in [-1, 1]^m$, we have

$$\underset{\substack{S \leftarrow (2\text{Bernoulli}(p)-1)^n, \\ T \leftarrow M(S), E}}{\mathbb{P}} \left[ S_i = 1 \wedge E_{M(s_{<i},1),M(s_{<i},-1)}(T) = 1 \big| S_{<i} = s_{<i}, T = t \right] \leq \frac{p \cdot e^\varepsilon}{p \cdot e^\varepsilon + 1 - p},$$

$$\underset{\substack{S \leftarrow (2\text{Bernoulli}(p)-1)^n, \\ T \leftarrow M(S), E}}{\mathbb{E}} \left[ E_{M(s_{<i},1),M(s_{<i},-1)}(T) \big| S_{\leq i} = (s_{<i}, 1) \right] \geq 1 - \delta,$$

$$\underset{\substack{S \leftarrow (2\text{Bernoulli}(p)-1)^n, \\ T \leftarrow M(S), E}}{\mathbb{E}} \left[ E_{M(s_{<i},1),M(s_{<i},-1)}(T) \big| S_{\leq i} = (s_{<i}, -1) \right] \geq 1 - \delta.$$

Symmetrically, for all $i \in [m]$, all $s_{<i} \in \{-1, 1\}^{i-1}$, and all $t \in [-1, 1]^m$, we have

$$\underset{\substack{S \leftarrow (2\text{Bernoulli}(p)-1)^n, \\ T \leftarrow M(S), E}}{\mathbb{P}} \left[ S_i = -1 \wedge E_{M(s_{<i},-1),M(s_{<i},1)}(T) = 1 \big| S_{<i} = s_{<i}, T = t \right] \leq \frac{(1-p) \cdot e^\varepsilon}{(1-p) \cdot e^\varepsilon + p},$$

$$\underset{\substack{S \leftarrow (2\text{Bernoulli}(p)-1)^n, \\ T \leftarrow M(S), E}}{\mathbb{E}} \left[ E_{M(s_{<i},-1),M(s_{<i},1)}(T) \big| S_{\leq i} = (s_{<i}, -1) \right] \geq 1 - \delta,$$

$$\underset{\substack{S \leftarrow (2\text{Bernoulli}(p)-1)^n, \\ T \leftarrow M(S), E}}{\mathbb{E}} \left[ E_{M(s_{<i},-1),M(s_{<i},1)}(T) \big| S_{\leq i} = (s_{<i}, 1) \right] \geq 1 - \delta.$$

For simplicity, we define a symmetric event: $E_P^Q(y) = E_Q^P(y) := E_{P,Q}(y) \cdot E_{Q,P}(y)$, where the internal randomnesses are again independent. Combining these, we have, for all $i \in [m]$, all $s_{<i} \in \{-1, 1\}^{i-1}$, and all $t \in [-1, 1]^m$,

$$\underset{\substack{S \leftarrow (2\text{Bernoulli}(p)-1)^n, \\ T \leftarrow M(S), E}}{\mathbb{P}} \left[ S_i = \text{sign}(T_i) \wedge E_{M(s_{<i},1)}^{M(s_{<i},-1)}(T) = 1 \big| T = t, S_{<i} = s_{<i} \right] \leq \begin{cases} \frac{p \cdot e^\varepsilon}{p \cdot e^\varepsilon + 1 - p} & \text{if } t_i > 0 \\ \frac{(1-p) \cdot e^\varepsilon}{(1-p) \cdot e^\varepsilon + p} & \text{if } t_i < 0 \end{cases}$$

and, for $b \in \{-1, 1\}$, we have

$$\underset{\substack{S \leftarrow (2\text{Bernoulli}(p)-1)^n, \\ T \leftarrow M(S), E}}{\mathbb{E}} \left[ E_{M(s_{<i},-1)}^{M(s_{<i},1)}(T) \big| S_{\leq i} = (s_{<i}, b) \right] \geq 1 - 2\delta.$$

For $k \in [m]$, $s \in \{-1, 1\}^m$, and $t \in [-1, 1]^m$, define

$$\widetilde{W}_k(s, t) := \sum_{i \in [k]} \max\{0, t_i \cdot s_i\} \cdot E_{M(s_{<i},1)}^{M(s_{<i},-1)}(t) = \sum_{i \in [k]} |t_i| \cdot \mathbb{I}[s_i = \text{sign}(t_i) \wedge E_{M(s_{<i},1)}^{M(s_{<i},-1)}(t) = 1]$$

and
$$\check{W}_k(t) = \sum_{i \in [k]} \check{S}_i(t) \cdot |t_i|,$$

where, for each $i \in [k]$ independently, $\check{S}(t)_i \leftarrow \mathsf{Bernoulli}(\frac{p \cdot e^\varepsilon}{p \cdot e^\varepsilon + 1 - p})$ if $t_i > 0$ and $\check{S}(t)_i \leftarrow \mathsf{Bernoulli}(\frac{(1-p) \cdot e^\varepsilon}{(1-p) \cdot e^\varepsilon + p})$ if $t_i < 0$.

By induction and Lemma 4.9, for any $k \in [m]$ and $t \in [-1, 1]^m$, the conditional distribution $(\widetilde{W}_k(S, t) | M(S) = t)$ where $S \leftarrow (2\mathsf{Bernoulli}(p) - 1)^m$ is stochastically dominated by $\check{W}_k(t)$.

For $s \in \{-1, 1\}^m$ and $t \in [-1, 1]^m$, define

$$F(s, t) := \sum_i^m \mathbb{I}\left[E_{M(s_{<i}, 1)}^{M(s_{<i}, -1)}(t) = 0\right],$$

so that

$$W_m(s, t) := \sum_{i \in [m]} \max\{0, t_i \cdot s_i\} \leq \widetilde{W}_m(s, t) + F(s, t).$$

Since the conditional distribution $(\widetilde{W}_k(S, t) | M(S) = t)$ where $S \leftarrow (2\mathsf{Bernoulli}(p) - 1)^m$ is stochastically dominated by $\check{W}_k(t)$, $W_m$ is stochastically dominated by the convolution $\check{W}_m(T) + F(S, T)$.

Finally $F(s, t)$ is supported on $\{0, 1, \cdots, m\}$ and

$$\mathbb{E}\left[F(s, t)\right] = \sum_i^m \mathbb{P}\left[E_{M(s_{<i}, 1)}^{M(s_{<i}, -1)}(T) = 0\right] \leq 2m \cdot \delta.$$

Since $\check{W}_m(T)$ does not depend on $S$, the input $S$ does not contribute to the dependence between $F(S, T)$ and $\check{W}_m(T)$, so we can elide this input in the statement – i.e., $F(T) = F(S, T)$ for $S$ drawn from an appropriate distribution. $\square$

Proposition 5.7 is rather unwieldy. It can be simplified by setting $p = \frac{1}{2}$ and identifying the optimal distribution $F(T)$, which yields Theorem 5.2.

*Proof of Theorem 5.2.* Let $M : \{-1, 1\}^m \to [-1, 1]^m$ satisfy $(\varepsilon, \delta)$-DP. Let $S \in \{-1, 1\}^m$ be uniformly random. Let $T = M(S) \in [-1, 1]^m$. Setting $p = \frac{1}{2}$ in Proposition 5.7 and averaging over $T$, we have, for all $v \in \mathbb{Z}$,

$$\mathop{\mathbb{P}}_{\substack{S \leftarrow \{-1,1\}^m \\ T \leftarrow M(S)}}\left[\sum_i^m \max\{0, T_i \cdot S_i\} \geq v\right] \leq \mathop{\mathbb{P}}_{\substack{S \leftarrow \{-1,1\}^m, T \leftarrow M(S), \\ \check{S} \leftarrow \mathsf{Bernoulli}\left(\frac{e^\varepsilon}{e^\varepsilon + 1}\right)^m, F}}\left[F(T) + \sum_i^m \check{S}_i \cdot |T_i| \geq v\right],$$

where $F$ is arbitrary – but independent from $\check{S}$ – except for the constraints that $F(T)$ is supported on $\{0, 1, \cdots, m\}$ and $\mathbb{E}\left[F(T)\right] \leq 2m \cdot \delta$.

Given these constraints, we can formulate finding the optimal distribution $F(t)$ for a given $t \in [-1, 1]^m$ and $v \in \mathbb{R}$ as a linear program:

$$\text{maximize} \quad \mathbb{P}_{\check{W},F}\left[\check{W}(t) + F(t) \geq v\right] = \sum_{i=0}^{m} \mathbb{P}_{F}[F(t) = i] \cdot \mathbb{P}_{\check{W}}\left[\check{W}(t) \geq v - i\right]$$

$$\text{subject to} \quad \mathbb{E}_{F}[F(t)] = \sum_{i=0}^{m} \mathbb{P}_{F}[F(t) = i] \cdot i \leq 2m \cdot \delta,$$

$$\sum_{i=0}^{m} \mathbb{P}_{F}[F(t) = i] = 1, \text{ and}$$

$$\mathbb{P}_{F}[F(t) = i] \geq 0 \quad \forall i \in \{0, 1, \cdots, m\},$$

where $\check{W}(t) := \sum_{i}^{m} \check{S}_i |t_i|$ for $\check{S} \leftarrow \mathsf{Bernoulli}\left(\frac{e^\varepsilon}{e^\varepsilon + 1}\right)^m$.

By strong duality, the linear program above has the same value as its dual:

$$\text{minimize} \quad 2m\delta\alpha + \beta$$

$$\text{subject to} \quad \alpha \cdot i + \beta \geq \mathbb{P}_{\check{W}}\left[\check{W}(t) \geq v - i\right] \quad \forall i \in \{0, 1, \cdots, m\},$$

$$\alpha \geq 0.$$

Any feasible solution to the dual gives an upper bound on the primal. So, in particular, we can use the solution given by

$$\beta = \mathbb{P}_{\check{W}^*}\left[\check{W}^* \geq v\right],$$

$$\alpha = \max\left(\{0\} \cup \left\{\frac{1}{i}\left(\mathbb{P}_{\check{W}^*}\left[\check{W}^* \geq v - i\right] - \beta\right) : i \in \{1, 2, \cdots, m\}\right\}\right),$$

where $\check{W}^*$ is a distribution on $\mathbb{R}$ that satisfies $\mathbb{P}_{\check{W}^*}\left[\check{W}^* \geq v - i\right] \geq \mathbb{P}_{\check{W}}\left[\check{W}(t) \geq v - i\right]$ for all $i \in \{0, 1, \cdots, m\}$ and all $t$ in the support of $T$. $\qquad \square$

Theorem 5.2 gives a worst-case bound in terms of $T$. Specifically, $\check{W}^*$ must uniformly bound $\check{W}(t)$ for all $t$ in the support of $T$. Proposition 5.7 is more general than this. Thus we give another corollary that allows us to have the bound adjust to $T$. In particular, this result allows the auditing procedure (Algorithm 1 or 3) to dynamically choose the number of guesses $r = k_+ + k_-$.

**Corollary 5.8** (Variant of Main Result). *Let* $M : \{-1, +1\}^m \to [-1, +1]^m$ *satisfy* $(\varepsilon, \delta)$-*DP. Let* $S \in \{-1, +1\}^m$ *be uniformly random. Let* $T = M(S) \in [-1, +1]^m$. *Then, for all* $\gamma \in [0, 1]$ *and* $\tau > 0$,

$$\mathbb{P}_{\substack{S \leftarrow \{-1,+1\}^m \\ T \leftarrow M(S)}}\left[\sum_{i}^{m} \max\{0, T_i \cdot S_i\} \geq g_{m,\varepsilon}(T, \gamma) + \tau\right] \leq \gamma + \frac{2m\delta}{\tau},$$

where $g_{m,\varepsilon} : [-1, +1]^m \times [0, 1] \to \mathbb{R}$ is an arbitrary function satisfying

$$\forall t \in [-1, 1]^m \; \forall \gamma \in [0, 1] \quad \mathbb{P}_{\check{S} \leftarrow \mathsf{Bernoulli}\left(\frac{e^\varepsilon}{e^\varepsilon+1}\right)^m}\left[\sum_i^m |t_i| \cdot \check{S}_i \geq g_{m,\varepsilon}(t, \gamma)\right] \leq \gamma.$$

*Proof.* Setting $p = \frac{1}{2}$ in Proposition 5.7 yields

$$\forall v \in \mathbb{R} \; \forall t \in [-1, +1]^m \quad \mathbb{P}_{\substack{S \leftarrow \{-1,+1\}^m \\ T \leftarrow M(S)}}\left[\sum_i^m \max\{0, T_i \cdot S_i\} \geq v \middle| T = t\right]$$

$$\leq \mathbb{P}_{\check{S} \leftarrow \mathsf{Bernoulli}\left(\frac{e^\varepsilon}{e^\varepsilon+1}\right), F}\left[F(t) + \sum_i^m |t_i| \cdot \check{S}_i \geq v\right],$$

where $F : [-1, 1]^m \to \{0, 1, \cdots, m\}$ satisfies $\mathbb{E}_{\substack{S \leftarrow \{-1,+1\}^m \\ T \leftarrow M(S), F}}[F(T)] \leq 2m\delta$.

By a union bound and Markov's inequality, we have, for all $t \in [-1, 1]^m$, all $\gamma \in [0, 1]$, and all $\tau > 0$,

$$\mathbb{P}_{\check{S} \leftarrow \mathsf{Bernoulli}\left(\frac{e^\varepsilon}{e^\varepsilon+1}\right), F}\left[F(t) + \sum_i^m |t_i| \cdot \check{S}_i \geq g_{m,\varepsilon}(t, \gamma) + \tau\right]$$

$$\leq \mathbb{P}_{\check{S} \leftarrow \mathsf{Bernoulli}\left(\frac{e^\varepsilon}{e^\varepsilon+1}\right), F}\left[\tau + \sum_i^m |t_i| \cdot \check{S}_i \geq g_{m,\varepsilon}(t, \gamma) + \tau\right] + \mathbb{P}_F[F(t) > \tau]$$

$$\leq \mathbb{P}_{\check{S} \leftarrow \mathsf{Bernoulli}\left(\frac{e^\varepsilon}{e^\varepsilon+1}\right), F}\left[\sum_i^m |t_i| \cdot \check{S}_i \geq g_{m,\varepsilon}(t, \gamma)\right] + \frac{\mathbb{E}_F[F(t)]}{\tau}$$

$$\leq \gamma + \frac{\mathbb{E}_F[F(t)]}{\tau}.$$

We combine inequalities, set $v = g_{m,\varepsilon}(t, \gamma) + \tau$, and average over $T$ to obtain

$$\forall \gamma \in [0, 1] \; \forall \tau > 0 \quad \mathbb{P}_{\substack{S \leftarrow \{-1,+1\}^m \\ T \leftarrow M(S)}}\left[\sum_i^m \max\{0, T_i \cdot S_i\} \geq g_{m,\varepsilon}(T, \gamma) + \tau\right]$$

$$\leq \gamma + \frac{\mathbb{E}_{\substack{S \leftarrow \{-1,+1\}^m \\ T \leftarrow M(S), F}}[F(T)]}{\tau}$$

$$\leq \gamma + \frac{2m\delta}{\tau}.$$

$\square$

---
**Algorithm 3** DP-SGD Auditor (Instantiation of Algorithm 1)
---
1: **Data:** $x \in \mathcal{X}^n$ consisting of $m$ auditing examples (a.k.a. canaries) and $n - m$ non-auditing examples.
2: **Parameters:** Number of examples to randomize $m$ for audit, number of positive $k_+$ and negative $k_-$ guesses `audit-type` (either `black-box` or `white-box`).
3: For $i \in [m]$ sample $S_i \in \{-1, +1\}$ independently with $\mathbb{E}[S_i] = 0$. Set $S_i = 1$ for all $i \in [n] \setminus [m]$.
4: Split $x$ into $x_{\text{IN}} \in \mathcal{X}^{n_{\text{IN}}}$ and $x_{\text{OUT}} \in \mathcal{X}^{n_{\text{OUT}}}$ according to $S$, where $n_{\text{IN}} + n_{\text{OUT}} = n$. Namely, if $S_i = 1$, then $x_i$ is in $x_{\text{IN}}$; and, if $S_i = -1$, then $x_i$ is in $x_{\text{OUT}}$.
5: Run DP-SGD (Algorithm 2) on input $x_{\text{IN}}$ with appropriate parameters.
6: Let $\ell$ be the number of iterations and let $f : \mathbb{R}^d \times \mathcal{X} \to \mathbb{R}$ be the loss.
7: Let $w^0, \cdots, w^\ell \in \mathbb{R}^d$ be the output of DP-SGD.
8: **if** `audit-type` = `black-box` **then**
9:     Define $\text{SCORE}(x_i, w^\ell) = \ell(w^0, x_i) - \ell(w^\ell, x_i)$ for all $i \in [m]$.
10:     Compute the vector of scores $Y = \left( \text{SCORE}(x_i, w^\ell) : i \in [m] \right) \in \mathbb{R}^m$.
11: **else if** `audit-type` = `white-box` **then**
12:     **procedure** $\text{SCORE}(x_*, w^1, \cdots, w^\ell)$
13:         **for** $t = 1, \cdots, \ell$ **do**
14:             Compute $g^t = \nabla_{w_{t-1}} \ell(w_{t-1}, x_*) \in \mathbb{R}^d$.
15:             Clip $\hat{g}^t = \min\left\{ 1, \frac{c}{\|g^t\|_2} \right\} \cdot g^t \in \mathbb{R}^d$.
16:             Let $v^t = \left\langle w^{t-1} - w^t, \hat{g}^t \right\rangle \in \mathbb{R}$.
17:         **end for**
18:         Return $\sum_{t=1}^{\ell} v^t \in \mathbb{R}$.
19:     **end procedure**
20:     Compute the vector of scores $Y = \left( \text{SCORE}(x_i, w^0, w^1, \cdots, w^\ell) : i \in [m] \right) \in \mathbb{R}^m$.
21: **end if**
22: Sort the scores $Y$. Let $T \in \{-1, 0, +1\}^m$ be $+1$ for the largest $k_+$ scores and $-1$ for the smallest $k_-$ scores.
23: (I.e., $T \in \{-1, 0, +1\}^m$ maximizes $\sum_i^m T_i \cdot Y_i$ subject to $\sum_i^m |T_i| = k_+ + k_-$ and $\sum_i^m T_i = k_+ - k_-$.)
24: **Return:** The vector $S \in \{-1, +1\}^m$ indicating the true selection and the guesses $T \in \{-1, 0, +1\}^m$.
---

# 6 Experiments

**Experiment Setup** Our contributions are focused on improved analysis of an existing privacy attack, and are therefore orthogonal to the design of an attack. As a result, we rely on the experimental setup of the recent auditing procedure of Nasr, Hayes, Steinke, Balle, Tramèr, Jagielski, Carlini, and Terzis [NHSBTJCT23].

We run DP-SGD on the CIFAR-10 dataset with Wide ResNet (WRN-16) [ZK16], we followed the experimental setup from Nasr et al. [NHSBTJCT23]. Our experiments reach 76% test accuracy at $(\varepsilon = 8, \delta = 10^{-5})$-DP, which is comparable with the state-of-the-art [DBHSB22]. Unless specified otherwise, all lower bounds are presented with 95% confidence. Following Nasr et al. [NHSBTJCT23], we refer to the setting where the adversary has access to all intermediate steps as "white-box" and when the adversary can only see the last iteration as "black-box." We experiment with both settings.

Algorithm 3 summarizes our approach for auditing DP-SGD. The results are converted into lower bounds on the privacy parameters using Theorem 5.2 / Corollary 5.4.

We also experiment with both the gradient and input attacks proposed by Nasr et al. [NHSBTJCT23]. In particular, for the gradient attack we use the strongest attack they

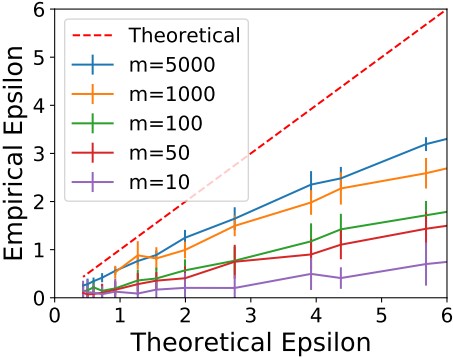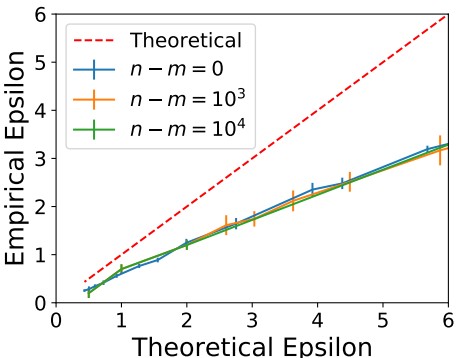

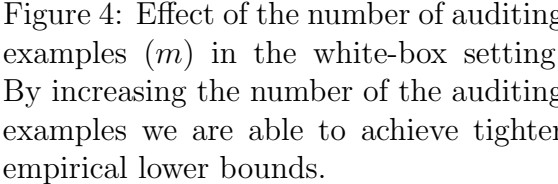

Figure 4: Effect of the number of auditing examples $(m)$ in the white-box setting. By increasing the number of the auditing examples we are able to achieve tighter empirical lower bounds.

Figure 5: Effect of the number of additional examples $(n - m)$ in the white-box setting. Importantly, adding additional examples does not impact the auditing results in the white-box setting.

proposed – the "Dirac canary" approach – which sets all gradients to zero except at a single random index. In our setting where we need to create multiple auditing examples (canaries) we make sure the indices selected in our experiments do not have any repetitions. To compute the score for gradient space attacks, we use the dot product between the gradient update and auditing gradient. When auditing in input space, we leverage two different types of injected examples as:

1. **Mislabeled example**: We select a random subset of the test set and randomly relabel them (ensuring the new label is not the same as the original label).

2. **In-distribution example**: We select a random subset of the test set.

For input space audits, we use the loss of the input example as the score. In our experiments we report the attack with the highest lower bound.

In our experiments, we evaluate different values of $k_+$ and $k_-$ and only report the highest auditing results. Since this is doing multiple hypothesis testing on the same data, we are reducing the confidence value of our results. However, this is commonly used in the previous works [ZBWTSRPNK22; MSS22] and can be easily improved by using a different set of observations to select the parameters for the auditing and another set of the data for the auditing itself (see also Corollary 5.8).

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

independent; we shouldn't expect them to be independent, but we also shouldn't expect them to be pathologically dependent in reality.

**Directions for further work:** Our work highlights several questions for further exploration:

- **Improved attacks:** Our experimental evaluation uses existing attack methods. Any improvements to membership inference attacks could be combined with our results to yield improved privacy auditing.

  One limitation of our attacks is that some examples may be "harder" than others and the scores we compute do not account for this. When we have many runs, we can account for the hardness of individual examples [CCNSTT22], but in our setting it is not obvious how to do this.

- **Algorithm-specific analyses:** Our methods are generic – they can be applied to essentially any DP algorithm. This is a strength, but there is also the possibility that we could obtain stronger results by exploiting the structure of specific algorithms. A

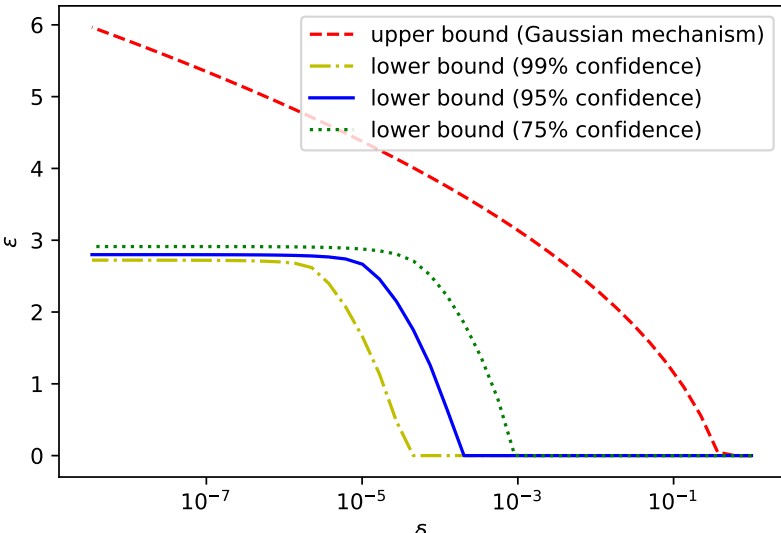

Figure 12: Comparison of upper and lower bounds for idealized setting with varying $\delta$. For each example $i \in [m]$, we release $S_i + \xi_i$, where $\xi_i \leftarrow \mathcal{N}(0,4)$ and $S_i \in \{-1,+1\}$ is independently uniformly random and indicates whether the sample is included/excluded. For the upper bound, we compute the exact $(4.38, 10^{-5})$-DP guarantee for the Gaussian mechanism (Lemma 4.5). For the lower bound, we plot the bound of Theorem 5.2 with 75%, 95%, and 99% confidence. We consider $m = 100,000$ randomized examples and $1,500$ guesses of which $1,429$ are correct. This corresponds to guessing $T_i = +1$ for the largest 750 scores, $T_i = -1$ for the smallest 750 scores, and $T_i = 0$ for the remaining 98,500 examples.

natural example of such structure is the iterative nature of DP-SGD. That is, we can view one run of DP-SGD as the composition of multiple independent DP algorithms which are run sequentially.

- **Multiple runs & multiple examples:** Our method performs auditing by including or excluding multiple examples in a single training run, while most prior work performs multiple training runs with a single example example included or excluded. Can we get the best of both worlds? If we use multiple examples and multiple runs, we should be able to get tighter results with fewer runs.

- **Other measures of privacy:** Our theoretical analysis is tailored to the standard definition of differential privacy. But there are other definitions of differential privacy such as Rényi DP. And, in particular, many of the upper bounds (e.g., Proposition 4.6) are stated in this language. Hence it would make sense for the lower bounds also to be stated in this language.

- **Beyond lower bounds:** Privacy auditing produces empirical lower bounds on the

**Algorithm 4** Pathological Algorithm

---

1: **Input:** $s \in \{-1, +1\}^m$
2: **Parameters:** $r \in [m]$, $\varepsilon, \delta \geq 0$, $\beta \in [0, 1]$. Assume $0 < m\delta \leq r\beta$.
3: Select $U \subset [m]$ of size $|U| = r$ uniformly at random.
4: Set $T_i = 0$ for all $i \notin U$.
5: Sample $X \leftarrow \mathsf{Bernoulli}(\beta)$.
6: **if** $X = 1$ **then**
7:      **for** $i \in U$ **do**
8:          Independently sample $T_i \in \{-1, +1\}$ with $\mathbb{P}\left[T_i = s_i\right] = \frac{m\delta}{r\beta} + \left(1 - \frac{m\delta}{r\beta}\right) \cdot \frac{e^\varepsilon}{e^\varepsilon + 1}$.
9:      **end for**
10: **else if** $X = 0$ **then**
11:      **for** $i \in U$ **do**
12:          Independently sample $T_i \in \{-1, +1\}$ with $\mathbb{P}\left[T_i = s_i\right] = \frac{e^\varepsilon}{e^\varepsilon + 1}$.
13:      **end for**
14: **end if**
15: **Output:** $T \in \{-1, 0, +1\}^m$.

---

privacy parameters. In contrast, mathematical analysis produces upper bounds. Both are necessarily conservative, which leaves a large gap between the upper and lower bounds. A natural question is to find some middle ground – an estimate which is neither a lower nor upper bound, but provides some meaningful estimate of the "true" privacy loss. However, it is unclear what kind of guarantee such an estimate should satisfy, or what interpretation the estimate should permit.

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

| [Tar08] | G. Tardos. "Optimal probabilistic fingerprint codes". In: *Journal of the ACM (JACM)* 55.2 (2008), pp. 1–24 (cit. on p. 5). |
| --- | --- |
| [TTSSJC22] | F. Tramer, A. Terzis, T. Steinke, S. Song, M. Jagielski, and N. Carlini. "Debugging differential privacy: A case study for privacy auditing". In: *arXiv preprint arXiv:2202.12219* (2022). URL: https://arxiv.org/abs/2202.12219 (cit. on p. 1). |
| [Vad17] | S. Vadhan. "The complexity of differential privacy". In: *Tutorials on the Foundations of Cryptography: Dedicated to Oded Goldreich* (2017), pp. 347–450. URL: https://privacytools.seas.harvard.edu/files/privacytools/files/manuscript_2016.pdf (cit. on p. 6). |
| [WBK19] | Y.-X. Wang, B. Balle, and S. P. Kasiviswanathan. "Subsampled rényi differential privacy and analytical moments accountant". In: *The 22nd International Conference on Artificial Intelligence and Statistics*. PMLR. 2019, pp. 1226–1235. URL: https://arxiv.org/abs/1808.00087 (cit. on p. 7). |
| [WBKBGGG22] | Y. Wen, A. Bansal, H. Kazemi, E. Borgnia, M. Goldblum, J. Geiping, and T. Goldstein. "Canary in a Coalmine: Better Membership Inference with Ensembled Adversarial Queries". In: *arXiv preprint arXiv:2210.10750* (2022) (cit. on p. 5). |
| [ZBWTSRPNK22] | S. Zanella-Béguelin, L. Wutschitz, S. Tople, A. Salem, V. Rühle, A. Paverd, M. Naseri, and B. Köpf. "Bayesian estimation of differential privacy". In: *arXiv preprint arXiv:2206.05199* (2022) (cit. on pp. 5, 25). |
| [ZDW22] | Y. Zhu, J. Dong, and Y.-X. Wang. "Optimal accounting of differential privacy via characteristic function". In: *International Conference on Artificial Intelligence and Statistics*. PMLR. 2022, pp. 4782–4817. URL: https://arxiv.org/abs/2106.08567 (cit. on p. 7). |
| [ZK16] | S. Zagoruyko and N. Komodakis. "Wide residual networks". In: *arXiv preprint arXiv:1605.07146* (2016) (cit. on p. 24). |

# A  Sampling a Fixed-Size Dataset

Our auditing framework considers randomly including or excluding $m$ examples independently. This means that the size of the dataset is random. This may be undesirable.

Fortunately, we can fix this, without changing our theoretical analysis. But it does require more examples and it requires changing the definition of DP to consider pairs of datasets differing by the replacement of one person's data, rather than the addition or removal of one person's data (cf. Remark 4.4).

Recall that our auditing framework starts with $m$ examples $x_1, \cdots, x_m$ and then samples $S \in \{-1, +1\}^m$ uniformly at random. Then each example $x_i$ is included in the dataset if $S_i = +1$ and excluded if $S_i = -1$. Thus flipping $S_i$ corresponds to adding or removing $x_i$.

Instead we can start with $2m$ examples $x_1, \cdots, x_{2m}$ and then sample $S \in \{-1, +1\}^m$ uniformly. Now, if $S_i = +1$, we include $x_{2i}$ in the dataset and, if $S_i = -1$, we include $x_{2i-1}$ instead. Thus flipping $S_i$ corresponds to replacing $x_{2i}$ with $x_{2i-1}$ or vice versa.

This alternative approach ensures that we always include $m$ out of the $2m$ examples – i.e., the dataset size is not random. This still fits the formalism of our theoretical analysis (§5). However, the DP guarantee of the algorithm being audited (e.g., DP-SGD) must now be with respect to replacement of one example, rather than addition or removal.[5] The auditor also needs to change slightly; rather than being given $x_i$ and needing to guess whether or not it is included in the datsets, the auditor is given both $x_{2i}$ and $x_{2i-1}$ and must guess which of the two is included.

# B  Generalization from Differential Privacy

Our analysis builds on the connection between DP and generalization [DFHPRR15b; DFH-PRR15a; BNSSSU16; FS17; JLNRSMS19]. We now extend our theoretical results (§5) to this setting. The main difference between our analysis in Section 5 and the prior work on DP and generalization is that we restrict to i.i.d. binary inputs with a uniform distribution, while prior work considers i.i.d. inputs from an arbitrary set with an arbitrary distribution. Thus the prior work is more general, but, as we now show, we can reduce the general case to the binary case.

**Theorem B.1** (DP implies Generalization). *Let $A : \mathcal{X}^n \to \mathcal{Y} \times [0,1]$ be $(\varepsilon, \delta)$-DP (with respect to replacement). Let $P$ be a distribution on $\mathcal{X}$. Let $q : \mathcal{Y} \times \mathcal{X} \to [0,1]$. For $x \in \mathcal{X}^n$ and $y \in \mathcal{Y}$, denote $q(y, x) = \frac{1}{n} \sum_i^n q(y, x_i) \in [0,1]$ and $q(y, P) = \mathbb{E}_{X \leftarrow P}[q(y, X)] \in [0,1]$. Then, for all $\gamma \geq \frac{3}{2}\eta \geq 0$, we have*

$$\Pr_{\substack{X \leftarrow P^n \\ Y \leftarrow A(X)}} [q(Y, X) - q(Y, P) \geq \gamma] \leq \begin{array}{l} \mathbb{P}\left[\check{W} \geq \frac{(1+\gamma-\frac{3}{2}\eta)n}{2}\right] + 2 \cdot e^{-n\eta^2/2} \\ + \max_{i \in [n]} \frac{2n\delta}{i} \mathbb{P}\left[\frac{(1+\gamma-\frac{3}{2}\eta)n}{2} > \check{W} \geq \frac{(1+\gamma-\frac{3}{2}\eta)n}{2} - i\right] \end{array},$$

*where $\check{W} \leftarrow \mathsf{Binomial}\left(n, \frac{e^\varepsilon}{e^\varepsilon+1}\right)$.*

The proof of Theorem B.1 relies on the following technical lemma. This is using what is known as the "ghost samples" symmetrization technique [SZ20, Footnote 2].

**Lemma B.2.** *Let $x^+, x^- \in \mathcal{X}^n$. For $s \in \{-1, +1\}^n$, define $x^s \in \mathcal{X}^n$ by $x_i^s = x_i^+$ if $s_i = +1$ and $x_i^s = x_i^-$ if $s_i = -1$. Let $A : \mathcal{X}^n \to \mathcal{Y}$ be $(\varepsilon, \delta)$-DP (for replacement). Let $q : \mathcal{Y} \times \mathcal{X} \to [0,1]$, and denote $q(y, x) = \frac{1}{n} \sum_i^n q(y, x_i) \in [0,1]$ for $y \in \mathcal{Y}$ and $x \in \mathcal{X}^n$.*

---

[5]By group privacy, $(\varepsilon, \delta)$-DP for addition or removal implies $(2\varepsilon, (e^\varepsilon + 1) \cdot \delta)$-DP for replacement.

*Let $S \in \{-1, +1\}$ be uniform. Then, for all $v, r \geq 0$,*

$$\mathop{\mathbb{P}}_{\substack{S \leftarrow \{-1,+1\}^n \\ Y \leftarrow A(x^S)}} \left[ q(Y, x^S) - q(Y, x^{-S}) \geq \frac{v}{n} \right]$$

$$\leq \mathbb{P}\left[ \check{W} \geq \frac{v - r + n}{2} \right] + \max_{i \in [n]} \frac{2n\delta}{i} \mathbb{P}\left[ \frac{v - r + n}{2} > \check{W} \geq \frac{v - r + n}{2} - i \right] + e^{-r^2/2n},$$

*where $\check{W} \leftarrow \mathsf{Binomial}\left(n, \frac{e^\varepsilon}{e^\varepsilon + 1}\right)$.*

*Proof.* Let $R : [-1, +1] \to \{-1, +1\}$ denote the randomized rounding function. I.e., $\mathbb{E}[R(x)] = x$ for all $x \in [-1, 1]$. We define $M : \{-1, +1\}^n \to \{-1, +1\}^n$ as follows. The inputs $x^+, x^- \in \mathcal{X}^n$ are "hardcoded" into $M$ and, for this analysis, we do not consider them private. Instead the input is $s \in \{-1, +1\}^n$. The algorithm $M(s)$ first runs $A(x^s)$ and then postprocesses the output using the hardcoded information. Specifically, given $A(s) = y$, the output $M(s) \in \{-1, +1\}^n$ has a product distribution with $M(s)_i = R(q(y, x_i^+) - q(y, x_i^-)) \in \{-1, +1\}$ for all $y \in \mathcal{Y}$ and all $i \in [n]$. That is, for each coordinate $i \in [n]$, we independently randomly round $q(y, x_i^+) - q(y, x_i^-) \in [-1, +1]$ to $\{-1, +1\}$, where $y$ is the output of $A(x^s)$. By postprocessing, $M$ is $(\varepsilon, \delta)$-DP. Thus we can apply Theorem 5.2 to $M$. We have, for all $r, v \geq 0$,

$$\mathop{\mathbb{P}}_{\substack{S \leftarrow \{-1,+1\}^n \\ Y \leftarrow A(x^S)}} \left[ q(Y, x^S) - q(Y, x^{-S}) \geq \frac{v}{n} \right]$$

$$= \mathop{\mathbb{P}}_{\substack{S \leftarrow \{-1,+1\}^n \\ Y \leftarrow A(x^S)}} \left[ \sum_i^n q(Y, x_i^S) - q(Y, x_i^{-S}) \geq v \right]$$

$$= \mathop{\mathbb{P}}_{\substack{S \leftarrow \{-1,+1\}^n \\ Y \leftarrow A(x^S)}} \left[ \sum_i^n (q(Y, x_i^+) - q(Y, x_i^-)) \cdot S_i \geq v \right]$$

$$= \mathop{\mathbb{P}}_{\substack{S \leftarrow \{-1,+1\}^n \\ Y \leftarrow A(x^S)}} \left[ \mathop{\mathbb{E}}_R \left[ \sum_i^n R(q(Y, x_i^+) - q(Y, x_i^-)) \cdot S_i \right] \geq v \right]$$

$$\leq \mathop{\mathbb{P}}_{\substack{S \leftarrow \{-1,+1\}^n \\ Y \leftarrow A(x^S), R}} \left[ \sum_i^n R(q(Y, x_i^+) - q(Y, x_i^-)) \cdot S_i \geq v - r \right] + e^{-r^2/2n} \qquad \text{(Hoeffding \& union)}$$

$$= \mathop{\mathbb{P}}_{\substack{S \leftarrow \{-1,+1\}^n \\ T \leftarrow M(S)}} \left[ \sum_i^n T_i \cdot S_i \geq v - r \right] + e^{-r^2/2n}$$

$$= \mathop{\mathbb{P}}_{\substack{S \leftarrow \{-1,+1\}^n \\ T \leftarrow M(S)}} \left[ \sum_i^n 2 \max\{0, T_i \cdot S_i\} - |T_i| \geq v - r \right] + e^{-r^2/2n} \qquad (S_i \in \{-1, +1\})$$

$$= \mathop{\mathbb{P}}_{\substack{S \leftarrow \{-1,+1\}^n \\ T \leftarrow M(S)}} \left[ \sum_i^n \max\{0, T_i \cdot S_i\} \geq \frac{v - r + n}{2} \right] + e^{-r^2/2n} \qquad (|T_i| = 1)$$

$$\leq \mathbb{P}\left[ \check{W} \geq \frac{v - r + n}{2} \right] + \max_{i \in [n]} \frac{2n\delta}{i} \mathbb{P}\left[ \frac{v - r + n}{2} > \check{W} \geq \frac{v - r + n}{2} - i \right] + e^{-r^2/2n},$$

where $\check{W} \leftarrow \mathsf{Binomial}\left(n, \frac{e^\varepsilon}{e^\varepsilon+1}\right)$. Note that Theorem 5.2 applies with any distribution $\check{W}^*$ satisfying

$$\forall v \in \mathbb{R} \quad \mathbb{P}\left[\check{W}^* > v\right] \geq \sup_{t \in \mathsf{support}(M(s))} \mathbb{P}_{\check{S} \leftarrow \mathsf{Bernoulli}\left(\frac{e^\varepsilon}{e^\varepsilon+1}\right)^n}\left[\sum_i^n \check{S}_i \cdot |t_i| > v\right].$$

If $t \in \mathsf{support}(M(s))$, then $|t_i| = 1$ for all $i \in [n]$, which implies $\check{W}$ satisfies this requirement. In the analysis above, we used Hoeffding's inequality to show that the sum of randomized roundings is close to (within $r$ of) the unrounded sum with high probability and we carry this failure probability $e^{-r^2/2n}$ into the final result. $\qquad\square$

**Proposition B.3.** *Let $A : \mathcal{X}^n \to \mathcal{Y}$ be $(\varepsilon, \delta)$-DP (with respect to replacement). Let $q : \mathcal{Y} \times \mathcal{X} \to [0,1]$, and denote $q(y,x) = \frac{1}{n}\sum_i^n q(y, x_i) \in [0,1]$ for $y \in \mathcal{Y}$ and $x \in \mathcal{X}^n$. Let $P$ be a distribution on $\mathcal{X}$. Then, for all $\gamma, \eta \geq 0$, we have*

$$\mathbb{P}_{\substack{X,\widetilde{X} \leftarrow P^n \\ Y \leftarrow A(X)}}\left[q(Y, X) - q(Y, \widetilde{X}) \geq \gamma\right]$$

$$\leq \mathbb{P}\left[\check{W} \geq \frac{(1+\gamma-\eta)n}{2}\right] + \max_{i \in [n]} \frac{2n\delta}{i}\mathbb{P}\left[\frac{(1+\gamma-\eta)n}{2} > \check{W} \geq \frac{(1+\gamma-\eta)n}{2} - i\right] + e^{-n\eta^2/2},$$

*where $\check{W} \leftarrow \mathsf{Binomial}\left(n, \frac{e^\varepsilon}{e^\varepsilon+1}\right)$.*

*Proof.* The proof relies on Lemma B.2, which considers $x^+, x^- \in \mathcal{X}^n$ to be fixed. We now average the lemma over these being i.i.d. samples from $P$, which gives

$$\mathbb{E}_{X^+,X^- \leftarrow P^n}\left[\mathbb{P}_{\substack{S \leftarrow \{-1,+1\}^n \\ Y \leftarrow A(X^S)}}\left[q(Y, X^S) - q(Y, X^{-S}) \geq \frac{v}{n}\right]\right]$$

$$\leq \mathbb{P}\left[\check{W} \geq \frac{v-r+n}{2}\right] + \max_{i \in [n]} \frac{2n\delta}{i}\mathbb{P}\left[\frac{v-r+n}{2} > \check{W} \geq \frac{v-r+n}{2} - i\right] + e^{-r^2/2n},$$

where $\check{W} \leftarrow \mathsf{Binomial}\left(n, \frac{e^\varepsilon}{e^\varepsilon+1}\right)$. Since the samples from $P$ are independent, the coordinates of $X^+$ and $X^-$ are interchangeable, so

$$\mathbb{P}_{\substack{X,\widetilde{X} \leftarrow P^n \\ Y \leftarrow A(X)}}\left[q(Y, X) - q(Y, \widetilde{X}) \geq \gamma\right] = \mathbb{E}_{X^+,X^- \leftarrow P^n}\left[\mathbb{P}_{\substack{S \leftarrow \{-1,+1\}^n \\ Y \leftarrow A(X^S)}}\left[q(Y, X^S) - q(Y, X^{-S}) \geq \frac{v}{n}\right]\right]$$

for $v = \gamma n \geq 0$. Setting $r = n\eta$ yields the result. $\qquad\square$

*Proof of Theorem B.1.* Let $X, \widetilde{X} \leftarrow P^n$ be two independent samples. Let $Y \leftarrow A(X)$. Let $\check{W} \leftarrow \mathsf{Binomial}\left(n, \frac{e^\varepsilon}{e^\varepsilon+1}\right)$. By Proposition B.3, for all $\gamma, \eta \geq 0$, we have

$$\mathbb{P}_{\substack{X,\widetilde{X} \leftarrow P^n \\ Y \leftarrow A(X)}}\left[q(Y, X) - q(Y, \widetilde{X}) \geq \gamma\right]$$

$$\leq \mathbb{P}\left[\check{W} \geq \frac{(1+\gamma-\eta)n}{2}\right] + \max_{i \in [n]} \frac{2n\delta}{i}\mathbb{P}\left[\frac{(1+\gamma-\eta)n}{2} > \check{W} \geq \frac{(1+\gamma-\eta)n}{2} - i\right] + e^{-n\eta^2/2},$$

By Hoeffding's inequality, for all $\eta \geq 0$, we have

$$\forall y \in \mathcal{Y} \qquad \mathbb{P}_{\widetilde{X} \leftarrow P^n} \left[ q(y, \widetilde{X}) - q(y, P) \geq \frac{\eta}{2} \right] \leq \exp(-n\eta^2/2).$$

By a union bound, for all $\gamma \geq \eta \geq 0$, we have

$$\mathbb{P}_{\substack{X \leftarrow P^n \\ Y \leftarrow A(X)}} [q(Y, X) - q(Y, P) \geq \gamma] \leq \begin{aligned} &\mathbb{P}_{\substack{X, \widetilde{X} \leftarrow P^n \\ Y \leftarrow A(X)}} \left[ q(Y, \widetilde{X}) - q(Y, P) \geq \gamma - \eta/2 \right] \\ &+ \mathbb{P}_{\substack{X, \widetilde{X} \leftarrow P^n \\ Y \leftarrow A(X)}} \left[ q(Y, X) - q(Y, \widetilde{X}) \geq \eta/2 \right] \end{aligned}.$$

Combining inequalities yields the result:

$$\mathbb{P}_{\substack{X \leftarrow P^n \\ Y \leftarrow A(X)}} [q(Y, X) - q(Y, P) \geq \gamma]$$

$$\leq \mathbb{P} \left[ \check{W} \geq \frac{(1 + \gamma - \eta/2 - \eta)n}{2} \right] + 2 \cdot e^{-n\eta^2/2}$$

$$+ \max_{i \in [n]} \frac{2n\delta}{i} \mathbb{P} \left[ \frac{(1 + \gamma - \eta/2 - \eta)n}{2} > \check{W} \geq \frac{(1 + \gamma - \eta/2 - \eta)n}{2} - i \right].$$

$\square$

## B.1 Comparison to Prior Work on DP & Generalization

We now briefly compare our results to the prior work on the connection between DP and generalization [DFHPRR15b; DFHPRR15a; BNSSSU16; FS17; JLNRSMS19]. We focus on the work of Jung, Ligett, Neel, Roth, Sharifi-Malvajerdi, and Shenfeld [JLNRSMS19] as it has the sharpest results in the literature.

Note that the prior work is focused on the setting of adaptive data analysis, while we are focused on the setting of auditing. This difference is mostly cosmetic, but there is a material difference when the prior results are applied to our setting: In addition to outputting guesses, the prior works assume that the algorithm outputs a differentially private estimate of the number of correct guesses. The guarantee then is that this differentially private estimate is close to the distributional average (i.e., only half of the guesses being correct). In contrast, for auditing, we want the true number of correct guesses to be close to the distributional average and don't produce a DP estimate. We can convert between these two settings using the triangle inequality.

Below we state the accuracy guarantee that we compare against, followed by a corollary of Theorem B.1 that applies the triangle inequality and a union bound to ensure that it is directly comparable.

**Theorem B.4** ([JLNRSMS19, Theorem 3.5]). *Let $A : \mathcal{X}^n \to \mathcal{Y} \times [0, 1]$ be $(\varepsilon, \delta)$-DP (with respect to replacement). Let $P$ be a distribution on $\mathcal{X}$. Let $q : \mathcal{Y} \times \mathcal{X} \to [0, 1]$. For $x \in \mathcal{X}^n$*

and $y \in \mathcal{Y}$, denote $q(y,x) = \frac{1}{n}\sum_i^n q(y,x_i) \in [0,1]$ and $q(y,P) = \underset{X \leftarrow P}{\mathbb{E}}[q(y,X)] \in [0,1]$. Suppose

$$\underset{\substack{X \leftarrow P^n \\ (Y,Z) \leftarrow A(X)}}{\mathbb{P}} [|Z - q(Y,X)| \geq \alpha] \leq \beta.$$

Then, for any $c, d > 0$, we have

$$\underset{\substack{X \leftarrow P^n \\ (Y,Z) \leftarrow A(X)}}{\mathbb{P}} [|Z - q(Y,P)| > \alpha + e^\varepsilon - 1 + c + 2d] \leq \frac{\beta}{c} + \frac{\delta}{d}. \tag{13}$$

**Corollary B.5** (Theorem B.1, triangle inequality, & union bound)**.** *Let $A : \mathcal{X}^n \to \mathcal{Y} \times [0,1]$ be $(\varepsilon,\delta)$-DP (with respect to replacement). Let $P$ be a distribution on $\mathcal{X}$. Let $q : \mathcal{Y} \times \mathcal{X} \to [0,1]$. For $x \in \mathcal{X}^n$ and $y \in \mathcal{Y}$, denote $q(y,x) = \frac{1}{n}\sum_i^n q(y,x_i) \in [0,1]$ and $q(y,P) = \underset{X \leftarrow P}{\mathbb{E}}[q(y,X)] \in [0,1]$. Suppose*

$$\underset{\substack{X \leftarrow P^n \\ (Y,Z) \leftarrow M(X)}}{\mathbb{P}} [|Z - q(Y,X)| \geq \alpha] \leq \beta.$$

*Then, for all $\gamma \geq \frac{3}{2}\eta \geq 0$, we have*

$$\underset{\substack{X \leftarrow P^n \\ (Y,Z) \leftarrow A(X)}}{\mathbb{P}} [|Z - q(Y,P)| \geq \alpha + \gamma] \leq \beta + \begin{array}{l} \mathbb{P}\left[\check{W} \geq \frac{(1+\gamma-\frac{3}{2}\eta)n}{2}\right] + 2 \cdot e^{-n\eta^2/2} \\ + \max_{i \in [n]} \frac{2n\delta}{i} \mathbb{P}\left[\frac{(1+\gamma-\frac{3}{2}\eta)n}{2} > \check{W} \geq \frac{(1+\gamma-\frac{3}{2}\eta)n}{2} - i\right] \end{array}, \tag{14}$$

*where $\check{W} \leftarrow \mathsf{Binomial}\left(n, \frac{e^\varepsilon}{e^\varepsilon+1}\right)$.*

Equations 13 and 14 are directly comparable, but it is not immediately obvious how they compare. By setting $\delta = 0$, $\gamma = \frac{e^\varepsilon-1}{e^\varepsilon+1} + c$, $\eta = \frac{2}{5}c$, and applying Hoeffding's inequality to $\check{W}$, we can simplify Equation 14 to

$$\underset{\substack{X \leftarrow P^n \\ (Y,Z) \leftarrow A(X)}}{\mathbb{P}} \left[|Z - q(Y,P)| \geq \alpha + \frac{e^\varepsilon - 1}{e^\varepsilon + 1} + c\right] \leq \beta + 3 \cdot e^{-n\frac{2}{25}c^2} \tag{15}$$

For comparison, setting $\delta = 0$ in Equation 13 gives

$$\underset{\substack{X \leftarrow P^n \\ (Y,Z) \leftarrow A(X)}}{\mathbb{P}} [|Z - q(Y,P)| > \alpha + e^\varepsilon - 1 + c] \leq \frac{\beta}{c}. \tag{16}$$

Now we can compare the results more easily. The $e^\varepsilon - 1$ term in the accuracy bound of Jung, Ligett, Neel, Roth, Sharifi-Malvajerdi, and Shenfeld [JLNRSMS19] is improved to $\frac{e^\varepsilon-1}{e^\varepsilon+1}$ in our result, which is an improvement by a factor of at least two. This is (arguably) the dominant term, so our result is a significant improvement. In particular, if $\varepsilon \geq \log 2$, then Equation 13 gives a vacuous bound (since the value of $q$ is always in $[0,1]$ anyway), while our bound can be non-vacuous for any value of $\varepsilon$ (as $\frac{e^\varepsilon-1}{e^\varepsilon+1} < 1$).

However, there is another term in the accuracy bound – i.e., $c$. The failure probability either has a $1/c$ *multiplicative* factor or a $3 \cdot e^{-n\frac{2}{25}c^2}$ *additive* factor. How these compare depends on the value of $\beta$. To give a concrete comparison, suppose $\varepsilon = 1/3$, $n = 2000$, $\beta = \delta = 10^{-5}$, and we want a final failure probability of 0.05; then Theorem B.4 gives an error guarantee of $\alpha + 0.397$, while Corollary B.5 gives $\alpha + 0.308$.

# C  Mutual Information Bounds from DP

Our framework for the theoretical analysis (§5) is inspired by that of Steinke and Zakynthinou [SZ20]. In this appendix, we use our analysis to also improve one of their results. Specifically, they show that if $M : \{-1, 1\}^m \to \mathcal{Y}$ satisfies $(\varepsilon, \delta)$-DP and $S \in \{-1, 1\}^m$ is uniformly random, then

$$I(S; M(S)) \leq (e^\varepsilon - 1 + \delta) \cdot m \cdot \log e, \tag{17}$$

where $I(\cdot\,; \cdot)$ denotes the mutual information.[6] Prior work [DFHPRR15a; BS16] showed that, if $M : \mathcal{X}^m \to \mathcal{Y}$ satisfies $(\varepsilon, 0)$-DP and $S \in \mathcal{X}^m$ has as product distribution, then

$$I(S; M(S)) \leq \frac{1}{2}\varepsilon^2 \cdot m \cdot \log e. \tag{18}$$

The latter result is numerically better than the former result, but only holds for pure DP. (The latter result is also not restricted to binary inputs. However, if we do not restrict the input at all, then it is not possible to prove bounds under approximate DP.)

We improve the bound to the following. If $M : \{-1, 1\}^m \to \mathcal{Y}$ satisfies $(\varepsilon, \delta)$-DP and $S \in \{-1, 1\}^m$ is uniformly random, then

$$I(S; M(S)) \leq \frac{1}{8}\varepsilon^2 \cdot m \cdot \log e + \delta \cdot m \cdot \log 2. \tag{19}$$

**Proposition C.1.** *Let $M : \{0, 1\}^n \to \mathcal{Y}$ satisfy $(\varepsilon, \delta)$-DP. Let $S \in \{0, 1\}^n$ be sampled from* Bernoulli$(p)^n$. *Then*

$$I(S; M(S)) \leq n\delta h(p) + n(1 - \delta)h\left(\frac{p \cdot e^\varepsilon + 1 - p}{e^\varepsilon + 1}\right) - n(1 - \delta) \cdot \left(\log(1 + e^{-\varepsilon}) + \frac{\log(e^\varepsilon)}{e^\varepsilon + 1}\right),$$

*where $h(p) := p \log(1/p) + (1 - p) \log(1/(1 - p))$ is the binary Entropy function.*

*In particular, if $p = \frac{1}{2}$, then*

$$I(S; M(S)) \leq n\delta \log 2 + n(1 - \delta) \cdot \left(\log 2 - \log(1 + e^{-\varepsilon}) - \frac{\log(e^\varepsilon)}{e^\varepsilon + 1}\right)$$

$$\leq n\delta \log 2 + n(1 - \delta)\frac{\varepsilon^2}{8} \log e.$$

*Proof.* We apply the chain rule and convexity of KL divergence [FS18, Lemma 3.7]:

$$I(S; M(S)) = \sum_i^n I(S_i; M(S)|S_{<i}) \leq \sum_i^n I(S_i; M(S)|S_{-i}).$$

---

[6]Throughout this paper we use natural logarithms (so $\log e = 1$), including when defining information-theoretic quantities like mutual information. However, it is common to use base-2 logarithms in information theory (i.e., $\log_2 e \approx 1.44$). To avoid confusion, the statements (outside proofs) in this section are stated in a redundant way so that they would be correct regardless of the base of the logarithm, as long as we are consistent.

Fix $i \in [n]$ and fix $s_{-i} \in \{0, 1\}^{n-1}$. Now we must analyze

$$
\begin{aligned}
I(S_i; M(S) | S_{-i} = s_{-i}) &= I(S_i; M(S_i, s_{-i})) \\
&= p\mathrm{D}_1\left(M(1, s_{-i}) \| p M(1, s_{-i}) + (1-p) M(0, s_{-i})\right) \\
&\quad + (1-p)\mathrm{D}_1\left(M(1, s_{-i}) \| p M(1, s_{-i}) + (1-p) M(0, s_{-i})\right) \\
&= p\mathrm{D}_1\left(Q_1 \| Q_p\right) + (1-p)\mathrm{D}_1\left(Q_0 \| Q_p\right),
\end{aligned}
$$

where $Q_t := t M(1, s_{-i}) + (1-t) M(0, s_{-i})$ for $t \in [0, 1]$.

Since $M$ is $(\varepsilon, \delta)$-DP, we have $Q_t(S) \le e^\varepsilon \cdot Q_{1-t}(S) + \delta$ for all measurable $S \subset \mathcal{Y}$ and $t \in \{0, 1\}$. Thus we can apply Lemma 5.5: There exist distributions $Q_0', Q_0'', Q_1', Q_1''$ on $\mathcal{Y}$ such that $Q_0 = (1-\delta) \cdot Q_0' + \delta \cdot Q_0''$ and $Q_1 = (1-\delta) \cdot Q_1' + \delta \cdot Q_1''$ and $e^{-\varepsilon} \cdot Q_0'(S) \le Q_1'(S) \le e^\varepsilon \cdot Q_0'(S)$ for all measurable $S \subset \mathcal{Y}$.

Define distributions

$$
R_0 := \frac{e^\varepsilon \cdot Q_0' - Q_1'}{e^\varepsilon - 1} \qquad \text{and} \qquad R_1 := \frac{e^\varepsilon \cdot Q_1' - Q_0'}{e^\varepsilon - 1},
$$

so that $Q_0' = \frac{e^\varepsilon \cdot R_0 + R_1}{e^\varepsilon + 1}$ and $Q_1' = \frac{e^\varepsilon \cdot R_1 + R_0}{e^\varepsilon + 1}$. Hence

$$
Q_0 = \frac{e^\varepsilon (1 - \delta)}{e^\varepsilon + 1} \cdot R_0 + \frac{1 - \delta}{e^\varepsilon + 1} \cdot R_1 + \delta \cdot Q_0''
$$

and

$$
Q_1 = \frac{e^\varepsilon (1 - \delta)}{e^\varepsilon + 1} \cdot R_1 + \frac{1 - \delta}{e^\varepsilon + 1} \cdot R_0 + \delta \cdot Q_1''.
$$

This decomposition (which was first used by Kairouz, Oh, and Viswanath [KOV15]) states that we can view $Q_{s_i} = M(s_i, s_{-i})$ as a postprocessing of an $(\varepsilon, \delta)$-DP randomized response on the bit $s_i$. That is, with probability $\delta$, we output the bit $s_i$ with a flag indicating certainty; with probability $\frac{e^\varepsilon (1-\delta)}{e^\varepsilon + 1}$, we output $s_i$ with an uncertain flag; and, with probability $\frac{1-\delta}{e^\varepsilon + 1}$, we output $1 - s_i$ with the uncertain flag. We can postprocess this to generate a sample from $Q_{s_i} = M(s_i, s_{-i})$ as follows. If we receive $b \in \{0, 1\}$ with the uncertain flag, then output a sample from $R_b$. If we receive $b \in \{0, 1\}$ with the certain flag, then output a sample from $Q_b''$.

To be formal, define two distributions on the set $[4] = \{1, 2, 3, 4\}$ by

$$
\widetilde{Q}_0 = \left( \frac{e^\varepsilon (1 - \delta)}{e^\varepsilon + 1}, \frac{1 - \delta}{e^\varepsilon + 1}, \delta, 0 \right),
$$

$$
\widetilde{Q}_1 = \left( \frac{1 - \delta}{e^\varepsilon + 1}, \frac{e^\varepsilon (1 - \delta)}{e^\varepsilon + 1}, 0, \delta \right).
$$

Define the a randomized postprocessing function $F : [4] \to \mathcal{Y}$ by $F(1) = R_0$, $F(2) = R_1$, $F(3) = Q_0''$, and $F(4) = Q_1''$. Then we have $F(\widetilde{Q}_0) = Q_0$ and $F(\widetilde{Q}_1) = Q_1$.

Now we use the postprocessing property (a.k.a. the data processing inequality):

$$
\begin{aligned}
I(S_i; M(S) | S_{-i} = s_{-i}) &= p\mathrm{D}_1\left(Q_1 \| Q_p\right) + (1-p)\mathrm{D}_1\left(Q_0 \| Q_p\right) \\
&\le p\mathrm{D}_1\left(\widetilde{Q}_1 \big\| \widetilde{Q}_p\right) + (1-p)\mathrm{D}_1\left(\widetilde{Q}_0 \big\| \widetilde{Q}_p\right).
\end{aligned}
$$

A tedious calculation now yields the bound:

$$p\mathrm{D}_1\left(\widetilde{Q}_1\middle\|\widetilde{Q}_p\right) + (1-p)\mathrm{D}_1\left(\widetilde{Q}_0\middle\|\widetilde{Q}_p\right)$$

$$= p\left(\frac{1-\delta}{e^\varepsilon+1}\log\left(\frac{\frac{1-\delta}{e^\varepsilon+1}}{p\frac{1-\delta}{e^\varepsilon+1}+(1-p)\frac{e^\varepsilon(1-\delta)}{e^\varepsilon+1}}\right) + \frac{e^\varepsilon(1-\delta)}{e^\varepsilon+1}\log\left(\frac{\frac{e^\varepsilon(1-\delta)}{e^\varepsilon+1}}{p\frac{e^\varepsilon(1-\delta)}{e^\varepsilon+1}+(1-p)\frac{1-\delta}{e^\varepsilon+1}}\right) + \delta\log\left(\frac{\delta}{p\delta}\right)\right)$$

$$+ (1-p)\left(\frac{e^\varepsilon(1-\delta)}{e^\varepsilon+1}\log\left(\frac{\frac{e^\varepsilon(1-\delta)}{e^\varepsilon+1}}{p\frac{1-\delta}{e^\varepsilon+1}+(1-p)\frac{e^\varepsilon(1-\delta)}{e^\varepsilon+1}}\right) + \frac{1-\delta}{e^\varepsilon+1}\log\left(\frac{\frac{1-\delta}{e^\varepsilon+1}}{p\frac{e^\varepsilon(1-\delta)}{e^\varepsilon+1}+(1-p)\frac{1-\delta}{e^\varepsilon+1}}\right) + \delta\log\left(\frac{\delta}{(1-p)\delta}\right)\right)$$

$$= p\frac{1-\delta}{e^\varepsilon+1}\left(\log\left(\frac{1}{p+(1-p)e^\varepsilon}\right) + e^\varepsilon\log\left(\frac{e^\varepsilon}{pe^\varepsilon+(1-p)}\right)\right)$$

$$+ (1-p)\frac{1-\delta}{e^\varepsilon+1}\left(e^\varepsilon\log\left(\frac{e^\varepsilon}{p+(1-p)e^\varepsilon}\right) + \log\left(\frac{1}{pe^\varepsilon+(1-p)}\right)\right)$$

$$+ \delta\left(p\log(1/p) + (1-p)\log(1/(1-p))\right)$$

$$= \frac{1-\delta}{e^\varepsilon+1}\left((p+(1-p)e^\varepsilon)\log\left(\frac{1}{p+(1-p)e^\varepsilon}\right) + (1-p)e^\varepsilon\varepsilon + (pe^\varepsilon+1-p)\log\left(\frac{1}{pe^\varepsilon+1-p}\right) + pe^\varepsilon\varepsilon\right)$$

$$+ \delta h(p)$$

$$= (1-\delta)\left(\frac{p+(1-p)e^\varepsilon}{e^\varepsilon+1}\log\left(\frac{e^\varepsilon+1}{p+(1-p)e^\varepsilon}\right) + \frac{pe^\varepsilon+1-p}{e^\varepsilon+1}\log\left(\frac{e^\varepsilon+1}{pe^\varepsilon+1-p}\right) - \log(e^\varepsilon+1) + \frac{e^\varepsilon\varepsilon}{e^\varepsilon+1}\right)$$

$$+ \delta h(p)$$

$$= (1-\delta)\left(h\left(\frac{p+(1-p)e^\varepsilon}{e^\varepsilon+1}\right) - \log(e^\varepsilon+1) + \frac{e^\varepsilon\varepsilon}{e^\varepsilon+1}\right) + \delta h(p)$$

$$= (1-\delta)\left(h\left(\frac{pe^\varepsilon+(1-p)}{e^\varepsilon+1}\right) - \log(1+e^{-\varepsilon}) - \frac{\varepsilon}{e^\varepsilon+1}\right) + \delta h(p).$$

Combining inequalities and summing over $i \in [n]$ yields the first part of the result. The final part of the result is the bound

$$\forall \varepsilon \geq 0 \quad g(\varepsilon) := \log 2 - \log(1 + e^{-\varepsilon}) - \frac{\varepsilon}{e^\varepsilon+1} \leq \frac{\varepsilon^2}{8},$$

which can be verified by showing that $g(0) = g'(0) = 0$ and $\forall \varepsilon \geq 0$ $g''(\varepsilon) \leq \frac{1}{4}$ (or by plotting it). $\qquad\square$

# D   Implementation of Theorem 5.2

On the next page is `Python` pseudocode implementing Corollary 5.4. Some example usage:

- Suppose the auditor correctly guesses $v = 75$ out of $m = r = 100$ examples, with no abstentions. We have $\frac{75}{100} = \frac{3}{4} = \frac{e^{\log 3}}{e^{\log 3}+1}$. So we would expect this to correspond roughly to $\varepsilon = \log 3 \approx 1.09$. Theorem 5.2 gives p-value of 0.553 for the null hypothesis $\varepsilon \leq \log 3$ and $\delta = 0$; to obtain this result call `p_value_DP_audit(100,100,75,math.log(3),0)` in the code below. If we want 95% confidence, we obtain the lower bound $\varepsilon \geq 0.702$

by calling `get_eps_audit(100,100,75,0,0.05)`. If we set $\delta = 10^{-4}$, we obtain the weaker lower bound $\varepsilon \geq 0.699$ by calling `get_eps_audit(100,100,75,1e-4,0.05)`.

- Suppose the auditor correctly guesses $v = 75$ out of $r = 100$ guesses, but with a total of $m = 1000$ examples. I.e., the auditor abstains on $m - r = 900$ examples. We obtain a lower bound of $\varepsilon \geq 0.673$ for $\delta = 10^{-4}$ and 95% confidence. (This is slightly weaker than the $\varepsilon \geq 0.699$ lower bound we get when there are no abstentions.) This is obtained by calling `get_eps_audit(1000,100,75,1e-4,0.05)`.

```
# m = number of examples, each included independently with probability 0.5
# r = number of guesses (i.e. excluding abstentions)
# v = number of correct guesses by auditor
# eps,delta = DP guarantee of null hypothesis
# output: p-value = probability of >=v correct guesses under null hypothesis
def p_value_DP_audit(m, r, v, eps, delta):
    assert 0 <= v <= r <= m
    assert eps >= 0
    assert 0 <= delta <= 1
    q = 1/(1+math.exp(-eps))  # accuracy of eps-DP randomized response
    beta = scipy.stats.binom.sf(v-1, r, q)   # = P[Binomial(r, q) >= v]
    alpha = 0
    sum = 0   # = P[v > Binomial(r, q) >= v - i]
    for i in range(1, v + 1):
        sum = sum + scipy.stats.binom.pmf(v - i, r, q)
        if sum > i * alpha:
          alpha = sum / i
    p = beta + alpha * delta * 2 * m
    return min(p, 1)

# m = number of examples, each included independently with probability 0.5
# r = number of guesses (i.e. excluding abstentions)
# v = number of correct guesses by auditor
# p = 1-confidence e.g. p=0.05 corresponds to 95%
# output: lower bound on eps i.e. algorithm is not (eps,delta)-DP
def get_eps_audit(m, r, v, delta, p):
    assert 0 <= v <= r <= m
    assert 0 <= delta <= 1
    assert 0 < p < 1
    eps_min = 0  # maintain p_value_DP(eps_min) < p
    eps_max = 1  # maintain p_value_DP(eps_max) >= p
    while p_value_DP_audit(m, r, v, eps_max, delta) < p: eps_max = eps_max + 1
    for _ in range(30):  # binary search
      eps = (eps_min + eps_max) / 2
      if p_value_DP_audit(m, r, v, eps, delta) < p:
        eps_min = eps
      else:
        eps_max = eps
    return eps_min
```