# OpenReview forum: "Privacy Auditing with One (1) Training Run"
_NeurIPS.cc/2023/Conference — NeurIPS 2023 oral_

### Official Review · Reviewer_U5hR · 2023-06-21

**Soundness:** 4 excellent
**Presentation:** 4 excellent
**Contribution:** 4 excellent
**Rating:** 9
**Confidence:** 4

**Summary:**

The paper improves the computational efficiency of auditing differentially private machine learning systems by connecting differential privacy and statistical generalization. The authors propose the first 1-round scheme compared to the standard solutions with hundreds of training rounds.

The auditing procedure operates as follows. 1) Randomly identify $m$ data points to either include or exclude. 2) Run the algorithm on the randomly selected dataset. 3) Given the algorithm's output, the auditor “guesses” whether or not each data point was included or excluded.

The main theoretical contribution is an improved analysis that is tailored to yield tight bounds. The auditing scheme requires minimal assumptions. The authors theoretically justified the improved efficiency of auditing via membership inference on multiple examples simultaneously. The paper shows that standard membership inference attacks can be used for auditing analysis, i.e., exploiting the parallelism of multiple independent data points in a single run of the algorithm in lieu of multiple independent runs.


In experiments, the authors audited DP-SGD training on a WideResNet model, trained on the CIFAR10 dataset across multiple configurations. The experiments contain both gradient and input attacks. The experimental results confirm the contributions claimed by the authors before.


**Strengths:**

+ The paper gives the first scheme of one-round privacy auditing, which is computationally efficient. This is a totally new perspective, jumping out of prior literature.
+ Theoretical foundation has been well built, i.e., Theorem 3.1. The formulation of privacy auditing bridges guess and privacy, which can be adopted in future instantiations/applications.
+ The experimental results are convincing, i.e., confirming the paper's contribution.
+ The writing is excellent. The authors clearly introduce the scheme, and meantime, show their deep insights and creative knowledge. At least, I feel I learned a lot in this paper.
+ Personally speaking, I think this paper will motivate many future works.


**Weaknesses:**

I did not find the weakness.

**Questions:**

The paper is self-contained. The details in the supplementary are very sufficient.

**Limitations:**

The authors studied the limitation comprehensively.

---

> ### Author Rebuttal · Authors · 2023-08-09
>
> Thank you very much for your review!

---

> > ### Comment · Reviewer_U5hR · 2023-08-16
> >
> > Thanks for the reply!
> >
> > I have read all contents on this page. I will keep my rating.

---

### Official Review · Reviewer_x3UA · 2023-07-06

**Soundness:** 4 excellent
**Presentation:** 4 excellent
**Contribution:** 3 good
**Rating:** 7
**Confidence:** 3

**Summary:**

The paper gives a simple version of a differential privacy (DP) auditing, and the proposed method is related to the recent works (G. Andrew, P. Kairouz, S. Oh, A. Oprea, H. B. McMahan, and V. Suriyaku- mar. “One-shot Empirical Privacy Estimation for Federated Learning”, 2023 and S. Zanella-Beguelin, L. Wutschitz, S. Tople, A. Salem, V. Ruhle, A. Paverd, M. Naseri, and B. Kopf. “Bayesian estimation of differential privacy”, 2022). There is also a theoretically rigorous analysis for the proposed method.

Andrew et al. (2023) work seems very much related (does also one shot auditing) however more heuristic. This paper provides confidence intervals for the epsilons, whereas it seems to be an open problem whether the empirical epsilons by Andrew et al. (2023) can be rigorously connected to the theoretical guarantees.

The idea in the proposed method is simple: certain amount of data (canaries) is held out for auditing such that each sample is randomly included in the training, and in the end the total number of correct guesses (whether sample was or wasn't in the training data) will give the empirical epsilons (high-probability lower bounds). This ratio can be connected to the theoretical guarantees via Theorem 3.1. The main result is the analysis of the method (Theorem 3.1). The experiments are very similar to those used in the recent auditing paper by Nasr, Hayes, Steinke, Balle, Tram`er, Jagielski, Carlini, and Terzis.

A pathological example in the shows the tightness of the epsilons give by this auditing method, and also includes some discussion about the looseness of the bounds in realistic situations.

**Strengths:**

- The idea is simple, the analysis sound and the paper is easy to read. All in all I think fits well to NeurIPS
- The analysis is tight in a sense that there are pathological DP algorithms for which the empirical epsilons are tight.
- Contribution seems clear: puts one shot auditing on a more rigorous footing (more rigorous than the previously proposed approaches)

**Weaknesses:**



- It remains a bit unclear from the paper, how does it compare to other auditing methods? It would be nice to see how it compares to e.g. to the empirical epsilons given in [NHSBTJCT23]. Especially in case of black box attacks there seems to be a big gap. Also would be interesting to see how the result compare to the results given by the one shot method of [AKOOMS23]. Even if the epsilons reported here are on a more rigorous footing than those of [AKOOMS23], I would guess that in practice one might end up using the method of [AKOOMS23] if it tends to give much more realistic epsilons.

- It still remains a bit unclear how useful this would be in practice. Especially in the black box setting the results look weak: already with 10000 samples (all used for auditing) or small amount of additional samples (not used for auditing), the empirical epsilons are really far from the theoretical ones.

- One small weakness is perhaps certain lack of originality: the main contribution is the mathematical analysis, the main ideas behind this auditing method seem to be laid out already in the recent works [ZBWTSRPNK22] and [AKOOMS23].

**Questions:**

In the white box setting, your experiments show that increasing number of data samples increases the epsilon lower bound. In Figure 3 you have Cifar-10 results up to $m=5000$ samples. Did you try for $m>5000$? How close can you get? Why not to report results for $m>5000$?

In the white box setting, where you insert the canaries and get the best results:  how expensive is the attack itself? I mean if you construct the canaries at each step, how expensive it is compared to the model training?

This approach is for auditing an ML system, and then it seems reasonable to use a small amount of samples and report the epsilon lower bounds. What if the protection gets really strong for realistic training data sizes, are those epsilons meaningful anymore? Would there be a way to make the method dataset/model specific?

Typo, p.5:
"First, we evaluate the effect of the number of the auditing example"

**Limitations:**

Yes, there is an extensive discussion in the end about the limitations of this approach. The extended version (supplements) has also a discussion about possible directions for future work.

---

> ### Author Rebuttal · Authors · 2023-08-09
>
> We thank the reviewer for their comments, which we respond to now.
>
> **On comparison with previous works:** Nasr et al. [NHSBTJCT23]  uses multiple runs and, as a result, they achieve tighter bounds than we do for the same algorithm; however, we are not sure if there is any meaningful comparison that we can do here as their method cannot analyze our setting and vice versa.
> Andrew et al. [AKOOMS23] propose an epsilon estimation technique which may not be a lower bound; their results in Table 1 show an estimate of epsilon that is larger that the provable epsilon upper bound. In the version that was available before the NeurIPS deadline their estimated epsilon actually significantly overestimated the true epsilon i.e. the theoretical upper bound was 50 and the estimate was 78. Given that their approach does not produce lower bounds on epsilon, a direct comparison is not very meaningful. An interesting direction for further work is to combine our methods with their approach.
>
> **On weakness of the black-box results:** We agree that in the black-box setting our results are weak, as is the case in all previous work. The black-box setting is very challenging, and there is a need to develop better attacks. We note that our analysis can then be applied to better attacks that are developed in the future.
>
> **On novelty:** The idea of using multiple canaries to audit DP mechanisms has been mentioned for years now; the first paper on auditing DP-SGD by Jagielski et al. [JUO'20] introduced multiple examples and used group privacy to analyze their results. The main novelty of the work is in creating a framework to theoretically analyze the setting and to provide a tight mathematical analysis.
>
> **Experiments with $m>5000$:** We didn’t experiment with more canaries due to memory limitations. In Figures 9 & 10 we simulated results for up to 1,000,000 and 100,000 examples, respectively.
>
> **Cost of the attack:** In the whitebox setting, creating canaries is very simple as they are simple gradient vectors that have large value on a single coordinate. The main cost is computing the score function, which requires comparing each gradient update vector with all of the canaries, which is a dot product between two large vectors. This cost can be reduced in future by only focusing on a specific layer in the network or small set of parameters. In the black-box setting, the canaries are simply examples and we only need to evaluate the final loss, which is much easier than computing inner products.
>
> **Would there be a way to make the method dataset/model specific?** The choice of auditing examples/canaries and the score function can be tailored to the dataset/model.

---

> > ### Comment · Reviewer_x3UA · 2023-08-16
> >
> > Thanks for the replies! I am keeping my score.

---

### Official Review · Reviewer_icCj · 2023-07-06

**Soundness:** 4 excellent
**Presentation:** 4 excellent
**Contribution:** 3 good
**Rating:** 7
**Confidence:** 4

**Summary:**

The authors propose a scheme for auditing differentially private machine learning models with a single training run (instead of thousands as have been used so far).

**Strengths:**

1. limitations clearly explained and illustrated
2. paper is really well structured (except for related work; see weaknesses)
3. important impact within DP auditing

**Weaknesses:**

1. It would help the reader if the related work was moved to before the empirical evaluation. Methods from the empirical evaluation are adapted from related work so more explanations earlier on might help with understanding
2. the comparison with related work falls a bit short. it would be nice to whether and when the heuristic proposed by MEMPST21 is not practically applicable. If they provide a better bound that holds true in most scenarios (even if there is no mathematical guarantee for it), it might be preferable to a loose bound with mathematical guarantee.

**Questions:**

1.  in ll 123 f., the authors say that the design of the attack is orthogonal to the analysis of the privacy attack. I wonder whether this claim is necessarily always true. Design methodology introduced for attacks with 1000s of runs might suffer from a high variance that could be maybe be highly detrimental for a single training run?
2. Do the authors believe that the methodology could be extended to multiple training runs to have a tradeoff between either 1000 or 1 runs?

**Limitations:**

the limitations have been adequately addressed and illustrated

---

> ### Author Rebuttal · Authors · 2023-08-09
>
> We thank the reviewer for their comments, particularly the suggestions regarding the related work section.
>
> **1. Line 123:** We will clarify that there may be overlap between the design of attacks and our analysis. But, conceptually, most auditing attack design considerations are the same in our setting and in the setting of prior work. That is, we are focused on creating examples that have a high effect on the training procedure and then detecting that effect. Nevertheless, we will make it clear in the text.
>
> **2. Multiple runs:** Extending our analysis to handle both multiple runs and multiple examples is a very interesting topic for future work and something that should be possible. However, we do want to emphasize that, for large models, training a single model is very expensive, and we think it is also important to improve the single model setting.

---

> > ### Comment · Reviewer_icCj · 2023-08-13
> >
> > I thank the authors for their response. After having read the rebuttal and the other reviewers' comments, I am more confident that this work constitutes a good paper hat should be accepted.

---

### Official Review · Reviewer_ZoZd · 2023-07-07

**Soundness:** 4 excellent
**Presentation:** 4 excellent
**Contribution:** 3 good
**Rating:** 7
**Confidence:** 4

**Summary:**

This paper presents a one shot approach for auditing privacy. Their approach is as follows: given n independent input samples and a (dp) algorithm to audit, they divide the $n$ input samples into two groups $X$ and $Y$ of size $m$ and $n-m$ respectively. Then they randomly select a partition of the first part $(X_1,X_2)$, and only use $X_1 \cup Y$ as the input of the dp algorithm. After that, they choose / design an application dependent score function that helps decide if each member of $X$ appeared in the input of the algorithm or not. The accuracy of this decision function implies a lower bound on the privacy of the algorithm.

Their main technical observation is that if $X$ is partitioned through a poisson sampling procedure, then running the auditing algorithm for one run is as good having $m$ independent runs. Their algorithm works both in the case where the final result of the algorithm is released and in the case where the intermediate steps are also released. In the end they decide whether the algorithm could be $(\varepsilon, \delta)$-dp or not by observing that the number correct guesses has to be less than $r \cdot e^\varepsilon / (e^\varepsilon+1) + O(\sqrt{r})$ with high probability, where $r< m$ is the total number of guesses.

**Strengths:**

The experiments for gradient space attack suggest that their lower bounds are within constant factor of the theoretical upper bounds, in both the case where the adversary has access to all of the model iterates and in the case where the adversary only has access to the final model.

The review of related work is very good and well written.

Their empirical lower bounds converge to analytical bounds in some cases if delta is sufficiently small and the number of examples goes to infinity.

**Weaknesses:**

Their approach does not seem to extend well to the case where delta is not very small.

In the setting where the adversary only can audit the input space (as opposed to the gradient space), their results do not seem to be very strong. This is important because in practice the algorithm has to be private with respect to the input space and not the gradient space.

**Questions:**

In line 51 to 57, the authors mention that such results were previously unattainable in the setting where only one model could be trained. On a conceptual level, what do you think was the roadblock that previous work could not overcome and is overcome in this work?

It's not clear to me in the experiments whether the "meaningful" metric to consider is $n-m$ or $n / m$.

There's no other auditing baseline presented to compare to in Figure 6. How do your results compare to previous work?

Your empirical results suggest that the lower bounds converge to analytical bounds in some cases if delta is sufficiently small and the number of examples go to infinity. Is there an analytical explanation for this?

Something that's a bit confusing to me is that in Figure 10, the empirical lower bound on epsilon decays after some point. Can you explain why this is happening?

**Limitations:**

The authors have discussed the limitations of their work adequately.

---

> ### Author Rebuttal · Authors · 2023-08-09
>
> Thank you for your time and comments. We respond below.
>
> **Not small delta:** Our approach works well for reasonable values of delta. E.g. delta=10^-3 in Figures 1 & 2. Handling larger delta is difficult even with multiple runs.
>
> **Conceptual roadblock:** The fact that multiple examples was previously considered as an unproven heuristic shows that people had the right intuition. The high-level key to analyzing this seems to be framing it in terms of generalization. The DP=>generalization literature provides the right toolkit to analyze multiple examples, but we still needed to do a lot of technical analysis to obtain tight results.
>
> **Comparisons for Figure 6:** Please note that none of the previous works can provide guarantees for the single run setting that we are considering in this work, as a result there are no previous works that we can directly compare to. To attempt a comparison to prior work, we tried auditing the Gaussian mechanism with a single example and multiple runs. To obtain results comparable to Figure 10 (i.e. eps>=2.8 when the sensitivity and standard deviation are equal), we require at least 100 runs using prior methods in comparason to a single run using our method.
>
> **Why n-m:** The main point that we want to mention in these experiments is that by increasing the number of non auditing examples the auditing becomes harder. Therefore, it doesn’t make much difference if we use m/n or n-m. Moreover, machine learning models have limited capacity and the number of examples seems to be more important than the ratio. For example as we see in Figure 7 by increasing the number of auditing example significantly we see worse auditing performance.
>
> **Convergence of lower bound to upper bound:** Figure 9 shows that our empirical lower bound does converge to the theoretical upper bound as the number of examples increases and accuracy is held constant. This simply demonstrates that our main theorem is tight. The gap between the bounds is due to statistical uncertainty, which decreases as we add more samples.
>
> **Lower bound decays in Figure 10:** This simulation shows the effect of abstentions. I.e., abstaining on some examples allows us to focus on the clearest examples, which results in higher accuracy. In Figure 10 we increase the number of guesses by making fewer abstentions, which results in lower accuracy. (In contrast, the accuracy in Figure 9 remains constant as we add more guesses.)

---

> > ### Comment · Reviewer_ZoZd · 2023-08-15
> >
> > I thank the authors for their response and answering my questions.
> >
> > After having read the comments of the other reviewers, the responses of the authors, and parts of the appendix, I have updated my review. Overall, I think this is a good and very well written paper that deserves to be in NeurIPS.

---

### Official Review · Reviewer_D1TM · 2023-07-11

**Soundness:** 4 excellent
**Presentation:** 3 good
**Contribution:** 3 good
**Rating:** 7
**Confidence:** 3

**Summary:**

The paper studies the problem of auditing differentially private machine learning systems. They propose a procedure which does so in one training run -- the key is the ability to include/exclude multiple data items in the run, as well as a novel analysis via leveraging connections between DP and generalization. They conduct experiments auditing DP-SGD obtaining meaningful empirical privacy lower bounds.

**Strengths:**

1. The topic of the paper is important and timely.  As the authors put, a privacy audit allows to assess the tightness of mathematical analysis and detect errors in analysis and implementation. As differentially private machine learning systems are being deployed increasingly, the research has potential to be impactful in these areas.

2. The authors make interesting contributions, with strong theoretical underpinnings which directly lead to improved results in practice. In my limited experience, such results aren't are rather rare and I greatly enjoyed this aspect of the paper. Moreover, the idea of membership inference with multiple inclusion/exclusion of data items, has been used as a heuristic in the past, so this paper gives a mathematical justification for the heuristic.

3. The paper is very well-written -- it is insightful and to the point. The paper also contains an extended discussion (Section 6) on the limitation of this work (and related approaches).

**Weaknesses:**

1. In some parts of the paper, I found the writing to be too dense. This is especially true for Section 6. I encourage the authors to revise this part.

2. While the authors provide extensive experimental results testing various aspects of the procedure, I found the section to be poorly organized. I encourage the authors to provide an overview of what is to come in the beginning of the experiments section.

**Questions:**

1. Why the particular choice of score function for white-box attacks? Did the authors try other choices before settling on this?
2. What is the intuition behind allowing abstention?

**Limitations:**

I greatly appreciate that the authors provided an extensive discussion of the limitations. However, as suggested above, the writing is too dense in these parts.

---

> ### Author Rebuttal · Authors · 2023-08-09
>
> Thank you for your time in reading and reviewing our submission!
>
> **Writing:** We will edit the paper, and especially Section 6 for clarity.
>
> **Why this score function:** This is the same score function as used in the prior work, previous works showed it achieves tight results. Our approach works with any score function and it is an interesting direction for further work to devise new score functions.
>
> **Why abstention:** As shown by Carlini et al. (2022), membership inference attacks frequently perform much better on some “hard” examples than on others. Abstaining allows us to focus on only those “hard” examples which are easiest to perform membership inference on. In addition, the hardness of the examples also has an element of chance due to the noise added for privacy; any given example may be harder or easier depending on how the noise happens to affect its score.

---

> > ### Comment · Reviewer_D1TM · 2023-08-21
> > **Thanks!**
> >
> > I thank the authors for providing clarifications. I am still positive about the paper and will maintain my score!

---

### Decision · Program_Chairs · 2023-09-21

**Decision:**

Accept (oral)

**Comment:**

The authors propose an efficient approach (that uses a single training run) for auditing private ML systems. All of the reviewers are very positive about this paper. This is one of the rare submission where the authors' (neat) theoretical contributions is expected to have practical significance too.